Resource

# The development of early human lymphatic vessels as characterized by lymphatic endothelial markers

Shoichiro Yamaguchi[1], Natsuki Minamide[1], Hiroshi Imai [ID][2], Tomoaki Ikeda[3], Masatoshi Watanabe[4], Kyoko Imanaka-Yoshida[1] & Kazuaki Maruyama [ID][1✉]

## Abstract

Lymphatic vessel development studies in mice and zebrafish models have demonstrated that lymphatic endothelial cells (LECs) predominantly differentiate from venous endothelial cells via the expression of the transcription factor Prox1. However, LECs can also be generated from undifferentiated mesoderm, suggesting potential diversity in their precursor cell origins depending on the organ or anatomical location. Despite these advances, recapitulating human lymphatic malformations in animal models has been difficult, and considering lymphatic vasculature function varies widely between species, analysis of development directly in humans is needed. Here, we examined early lymphatic development in humans by analyzing the histology of 31 embryos and three 9-week-old fetuses. We found that human embryonic cardinal veins, which converged to form initial lymph sacs, produce Prox1-expressing LECs. Furthermore, we describe the lymphatic vessel development in various organs and observe organ-specific differences. These characterizations of the early development of human lymphatic vessels should help to better understand the evolution and phylogenetic relationships of lymphatic systems, and their roles in human disease.

**Keywords** Cellular Origin of Lymphatic Endothelial Cells; Human Embryos; Lymphatic Vessel Development
**Subject Categories** Development; Methods & Resources; Vascular Biology & Angiogenesis

## Introduction

The lymphatic vascular system is essential for fluid transportation, immune reaction and lipids absorption. Lymphatic vessels are also important for various pathophysiological conditions (Oliver et al, 2020; Alitalo, 2011; Maruyama and Imanaka-Yoshida, 2022), including myocardial infarction (Maruyama et al, 2021; Henri et al, 2016; Klotz et al, 2015; Matsui et al, 2023), encephalitis (Hsu et al,

2019), and Alzheimer's disease (Mesquita et al, 2018). The development of lymphatic vessels has long been a subject of debate. Early studies conducted in the 1900s proposed that initial lymphatic vessels (lymph sacs) originate from venous endothelial cells and the entire lymphatic system subsequently proliferates from these structures into adjacent tissues and organs (Sabin, 1902). An alternative theory suggested that lymph sacs originate from undifferentiated mesodermal cells and subsequently establish connections between jugular veins (Huntington and McClure, 1910). An anatomical study on lymphatic development using human embryos was conducted by Putte in 1975 (van der Putte, 1975). However, at that time, lymphatic markers had not been identified yet. As a result, it did not provide comprehensive detailed observations.

In recent years, progress in genetic analysis and the development of specific markers has significantly enhanced our understanding. The transcription factor Prospero homeobox protein 1 (Prox1) (Wigle and Oliver, 1999), a master regulator of lymphatic endothelial cell (LEC) differentiation, and the Vascular Endothelial Growth Factor C (VEGF-C)-VEGF Receptor 3 (VEGFR3) axis (Mustonen and Alitalo, 1995; Joukov et al, 1996), which is crucial for LECs proliferation, are essential molecules of the lymphatic vessel development. Genetic lineage analysis using mouse embryos with *Prox1-CreERT2* has clarified that LECs primarily originate from the cardinal veins (Srinivasan et al, 2007). Furthermore, we have recently elucidated that in mice, LECs not only derived from venous endothelial cells, but in the head and neck region and the mediastinum, they may originate from evolutionarily conserved cardiopharyngeal mesoderm, which also generates musculatures and connective tissues in these regions (Maruyama et al, 2022). It has also been reported that the progenitors of lymphatic vessels may differ among organs, such as the heart (Klotz et al, 2015; Lioux et al, 2020; Stone and Stainier, 2019), mesentery (Stanczuk et al, 2015), and skin (Pichol-Thievend et al, 2018; Martinez-Corral et al, 2015).

Lymphatic malformations are one of the most common vascular disorders that occur in approximately one in 2000–4000 births (Zenner et al, 2019). Lymphatic malformations tend to occur from the head and neck region to the mediastinum and are considered to be caused by abnormalities in the developmental process. Although mouse models of lymphatic malformations have been reported, it has been challenging to reproduce the anatomical characteristics of humans in mice (Mäkinen et al, 2021). This might be because the

---

[1]Department of Pathology and Matrix Biology, Graduate School of Medicine, Mie University, 2-174 Edobashi, Tsu, Mie 514-0001, Japan. [2]Pathology Division, Mie University School of Medicine, 2-174 Edobashi, Tsu, Mie 514-0001, Japan. [3]Department of Obstetrics and Gynecology, Mie University School of Medicine, 2-174 Edobashi, Tsu, Mie 514-0001, Japan. [4]Department of Oncologic Pathology, Graduate School of Medicine, Mie University, 2-174 Edobashi, Tsu, Mie 514-0001, Japan. ✉E-mail: k-maruyama0608@med.mie-u.ac.jp

 

lymphatic development in humans differs from that in the mice, but the exact cause remains poorly understood. Furthermore, each species possesses unique function and anatomy of the lymphatic system. For example, in tuna, the lymphatic vessels play a crucial role in moving the fins (Pavlov et al, 2017). In amphibians and reptiles, there are lymphatic hearts, which are part of lymphatic vessels that contract autonomously (Banda et al, 2023). While in birds, lymphatic hearts are once formed in embryonic periods, they disappear as the development progresses (Budras et al, 1987). Thus, the development, function, and anatomy of the lymphatic vessels display variations between species, and the knowledge obtained from mice does not necessarily apply to humans. Therefore, as a fundamental basis for studying human lymphatic diseases, it is crucial to understand lymphatic vessel development in humans.

In this study, we attempted to examine lymphatic vessel development in humans using 31 embryos and 3 fetuses at the 9th week of gestation (GW). We found that in human embryos, Prox1 is expressed in the cardinal veins at Carnegie stage (CS)12, initiating the emergence of LECs. Subsequently, LECs budding from the cardinal veins begin to express VEGFR3 at CS13. LYVE1 and PDPN are expressed in lymph sacs at CS16. A valve structure is formed on the lymphatic side of the junction between the lymph sac and the cardinal veins at CS18, creating a boundary between the cardinal veins and the lymphatic vessels. On the other hand, when observing in each organ (heart, lower jaw, lungs, mesentery, kidney, and thoracic duct), the development of lymphatic vessels varies.

Our study is the first to clarify the lymphatic vessel development in humans. LECs are preliminarily derived from embryonic veins. On the other hand, the process of lymphatic development in each organ is spatiotemporally diverse, which may be due to differences in their developmental processes or cellular origins. These findings indicate that the development of lymphatic vessels is similar between humans and other species. Our research offers essential insights into the evolution and phylogeny of lymphatic vessels, and may also illuminate the pathogenesis of lymphatic-related diseases, which include lymphedema, obesity, cardiovascular disorders, Crohn's disease, and congenital lymphatic disease, such as lymphatic malformation.

# Results

## Evaluation of marker expression in human fetal lymphatic endothelial cells

To explore the development of human lymphatic vessels, we tested antibodies against Prox1, VEGFR3, and lymphatic vessel endothelial hyaluronan receptor-1 (LYVE1), all of which are commonly employed in the study of murine lymphatic vessel development. In addition, we incorporated D2-40, a monoclonal antibody that specifically interacts with the glycoprotein Podoplanin (PDPN). This antibody is frequently used in human pathological diagnoses for staining to adult lymphatic vessels. We also employed platelet endothelial cell adhesion molecule-1 (PECAM) antibodies as a marker for endothelial cells. By GW9, we confirmed that most organs had developed sufficiently to resemble those found in adults. We proceeded with the fluorescent immunostaining and enzyme-antibody method, followed by 3,3′-diaminobenzidine (DAB) color development in this GW9 fetus and verified the expression of all marker proteins in LECs within jugular lymph sacs (Fig. EV1A–H).

In addition, the specificity of the staining was confirmed with controls using only the secondary antibodies (Fig. EV1I–L). From these observations, it became evident that these molecules are also expressed in lymphatic vessels in human fetuses.

## The emergence of lymphatic endothelial cells begins in the embryonic cardinal veins in human embryos

Next, we analyzed the development of lymphatic vessels using embryos at various stages. In CS11, when the formation of the precardinal vein occurred, we could not identify Prox1 expression in the precardinal vein (Fig. EV2A–C'). At CS12, we performed immunostaining using PECAM, Prox1, and Chicken ovalbumin upstream promoter–transcription factor II (COUP-TFII) antibodies. COUP-TFII, a member of the nuclear receptor superfamily, is necessary for the activation of Prox1 in the cardinal veins in mice (Yamazaki et al, 2009; Srinivasan et al, 2010). We identified Prox1$^+$/PECAM$^+$/COUP-TF2$^+$ and Prox1$^+$/PECAM$^+$/COUP-TF2$^-$ LECs in and around the anterior cardinal veins (ACVs) (Fig. 1A–I). At this stage, LYVE1 expression was not observed in the ACVs or its surroundings, whereas VEGFR3 was expressed in the ACVs (Fig. 1C,D,G,H). By CS13, Prox1$^+$/PECAM$^+$/Coup-TF2$^+$, or Prox1$^+$/PECAM$^+$/COUP-TF2$^-$ LECs formed capillary lymphatics extending towards the posterior of the body (Fig. 1J–K'''). While these capillary lymphatics expressed VEGFR3 (Fig. 1L–M'''), they did not express LYVE1 (Fig. 1N–O''').

We previously reported that the lymphatic vessels in the head and neck region originate from *Islet1* (*Isl1*)$^+$ LECs located in the pharyngeal arch mesodermal regions(Maruyama et al, 2019, 2022). In addition, it was suggested that these LECs may be derived from multipotent Isl1$^+$/Flk1$^+$ cardiovascular progenitors(Milgrom-Hoffman et al, 2011). Based on these findings, we performed immunostaining of Isl1, Flk1, PECAM, and Prox1 to identify LECs and their progenitor cells originating from the pharyngeal arches. We did not detect any Isl1$^+$ endothelial cells in and around the cardinal veins (Fig. 1P–Q'''). Although we detected a small number of Flk1$^+$/Isl1$^+$/PECAM$^-$ cells within the second pharyngeal arch mesoderm, we could not find any cells that expressed both Prox1 and Isl1 (Fig. 1R–U''').

At CS14, we observed more Flk1$^+$/Is1$^+$/PECAM$^-$ cells in the second pharyngeal arch (Fig. EV3A–C'''). However, we did not find any cells co-expressed Isl1 and Prox1 (Fig. EV3D–E'''). Around the ACVs, LECs partially formed luminal structures at CS14 (Fig. EV3F–J). In the CS15 embryo, we identified a few Prox1$^+$ cells and PECAM$^+$ cells within the second pharyngeal arch (Fig. EV3K,K',P,P'). However, due to poor fixation and excessive autofluorescence in multi-fluorescence staining, we were unable to determine whether PECAM and Prox1 were co-expressed in the pharyngeal arches. VEGFR3$^+$ cells were present in the second pharyngeal arch, but for the same reason, we could not determine whether they were expressed in LECs (Fig. EV3M,M'). The Expression of LYVE1 and PDPN was not observed in the pharyngeal arches (Fig. EV3N–O'). At CS16, cells co-expressing Prox1 and Isl1 were not observed in the lower jaw or the cardiac outflow tract regions (Fig. EV3Q–S'''). In addition, at GW9, Flk1 expression was detected in the cervical lymph sac, but Isl1 expression was not (Fig. EV1M,N).

In summary, LECs arise from the ACVs at CS12. We identified Flk1$^+$/Isl1$^+$/PECAM$^-$ cells in the second pharyngeal arch, but we were unable to identify any cells co-expressing Isl1 and Prox1 at CS13, CS14, and CS16. Therefore, we have concluded that LECs first originate from the ACVs in human embryos.

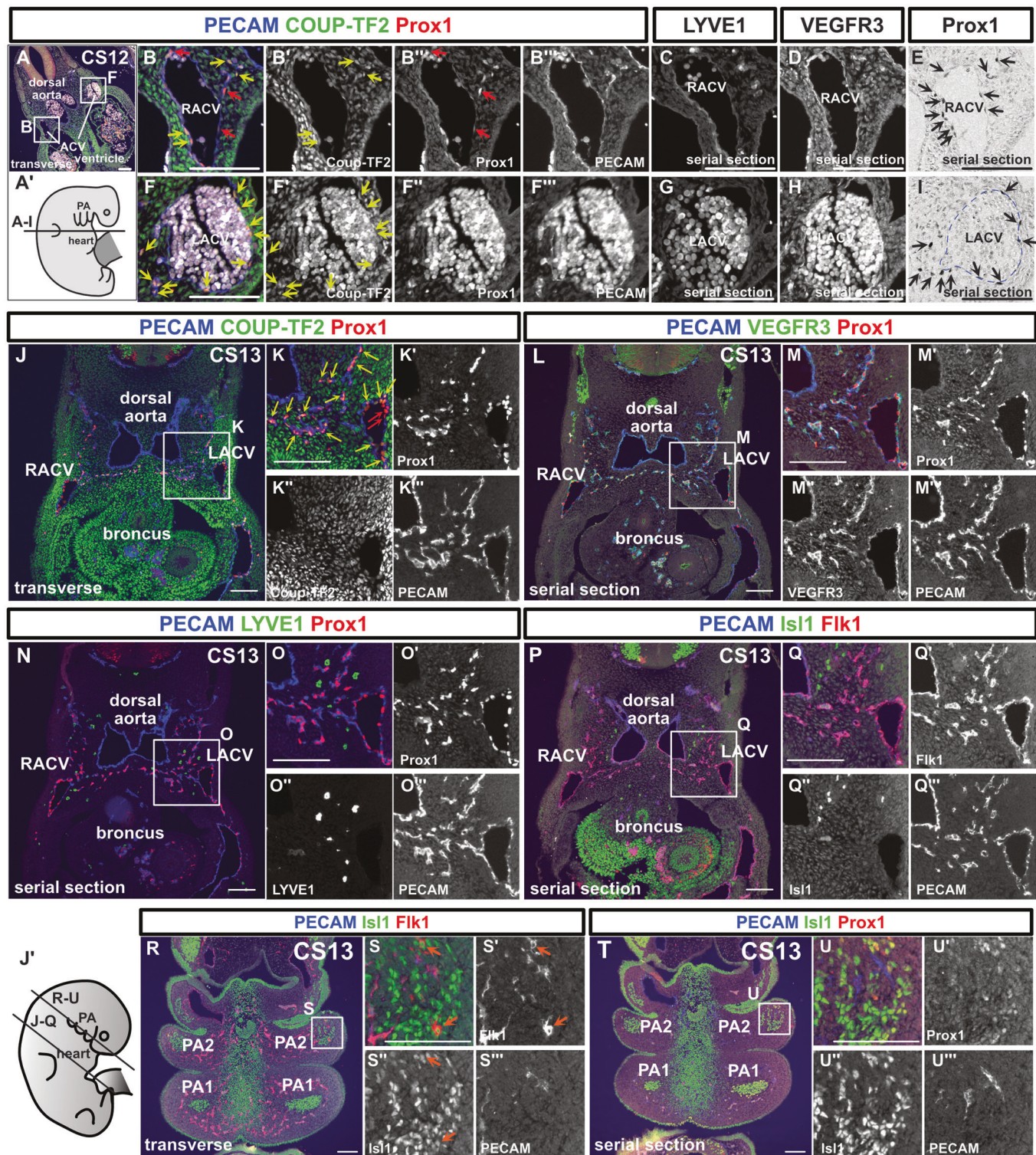

## Lymphovenous valves are formed between lymph sacs and the cardinal veins

At CS16, the large initial lymphatic vessels (lymph sacs) around the ACVs were formed, and they expressed LYVE1 and PDPN as well as VEGFR3 and Prox1 (Fig. 2A–D). By CS18, the connection between the ACVs and lymph sacs was observed and on lymph sacs side, valve-like protrusions began to appear (Fig. 2E–M). In the junction where the cardinal veins and lymphatic sacs merged, no VEGFR3, Prox1, LYVE1, or PDPN expression was seen on the venous side, clearly delineating the boundary between the lymphatic vessels and veins (Fig. 2F–L). Notably, while the venous

**Figure 1. Lymphatic endothelial cells originate from the cardinal veins in human embryos.**

(A–I) Immunostaining of transverse sections using the indicated antibodies in a CS12 embryo and schematic representation of the locations of the sections; We detected Prox1$^+$/PECAM$^+$/Coup-TF2$^-$ cells (red arrows) and Prox1$^+$/PECAM$^-$/Coup-TF2$^+$ cells (yellow arrows: 78.9% of PECAM$^+$/Prox1$^+$ cells were Coup-TF2$^+$, $n = 1$) in and around the ACVs. On average, there were 18.25 nuclei per cross-section of the CV; of these, 4.5 were Prox1$^-$/PECAM$^+$ blood endothelial cells (BECs), and 13.75 were Prox1$^+$/PECAM$^+$ LECs. Therefore, BECs constituted 24.7%, and LECs constituted 75.3%. There were an average of 9.75 Prox1$^+$/PECAM$^+$ cells located externally to the CV. (E, I) Enzyme-antibody method; Prox1$^+$ cells were present around the ACVs (black arrows). These Prox1$^+$/PECAM$^+$ cells were localized around the cardinal veins in the CS12 embryos. (J–U''') Fluorescent immunostaining of transverse sections using the indicated antibodies in a CS13 embryo and a schematic representation of the positions of the sections; The PECAM$^+$/Prox1$^+$ LECs budding from the ACVs consisted of Coup-TF2-expressing LECs (yellow arrows: 82.8 ± 2.18% of the PECAM$^+$/Prox1$^+$ cells were Coup-TF2$^+$, $n = 2$) and Coup-TF2$^-$ LECs (red arrows). PECAM$^-$/Isl1$^+$/Flk1$^+$ cardiovascular progenitor cells were observed in the pharyngeal arch mesoderm (orange arrow). CV cardinal vein, RACV right anterior cardinal vein, LACV left anterior cardinal vein, PA1 first pharyngeal arch, PA2 second pharyngeal arch. Scale bar, 100 µm (A–U). Source data are available online for this figure.

side maintained a smooth lumen, the lymphatic side appeared distorted, facilitating morphological distinction (Fig. 2E–M).

Similar findings were observed at stages CS19 and 21. At the junction between jugular lymph sacs and the jugular vein, lymphovenous valves (LVVs) were formed, and VEGFR3 expression was visible on the lymph sac side but absent from the venous side (Fig. 2N–Q'''). On both the lymph sac and venous sides, the valve leaflets were covered by Prox1$^+$ LVV endothelial cells (Fig. 2N–Q'''). The LVVs, which were initially irregularly shaped at CS18, gradually assumed a regular form as development progressed; i.e., by GW9 a bicuspid valve had developed (Fig. 2R–R''').

## The development of lymphatic vessels varies among organs

Consequently, we conducted an analysis of lymphatic vessel development across various organs.

### The cardiac lymphatic vessels
At CS16, capillary lymphatic vessels and isolated LECs in the wall of the aorta were observed (Fig. 3A). This lymphatic network was composed of PECAM$^+$/Prox1$^+$/VEGFR3$^+$ or PECAM$^+$/Prox1$^+$/VEGFR3$^-$ LECs, which did not express LYVE1 or PDPN (Fig. 3A–E). As development progressed, tubular lymphatic vessels were identifiable at CS23 (Fig. 3F–O). Throughout this process, the initially mesh-like capillary lymphatics undergo progressive remodeling to establish lumen-bearing vessels. Consequently, while the density of LECs per unit area remains relatively stable, there is an increase in the number of lymphatic vessels possessing distinct luminal structures (Fig. 3N,O). These lymphatic vessels expressed LYVE1 and PDPN at CS23 and GW9 (Figs. 3L,M and EV4X',Y'). By GW9, lymphatic vessels were identifiable on the epicardial side (Fig. 3P,Q). Lymphatic vessels were also distributed around the coronary arteries (Fig. 3P–R). No lymphatic vessels were seen in the myocardium or endocardium during the examined periods (Fig. 3S).

### Lung lymphatic vessels
At stage CS16, only a small number of PECAM$^+$/Prox1$^+$/VEGFR3$^+$ LECs could be detected in the lung parenchyma (Fig. 4A). However, as development progress increases in the number of LECs and luminal lymphatic vessels were evident (Fig. 4A–J). At CS23, lymphatic vessels were distributed around the trachea, which became apparent by GW9 (Fig. 4G,H). PDPN and LYVE1 expression were detected in the lymphatic vessels in the lung at GW9 (Fig. EV4X,Y).

### Mesenteric and intestinal lymphatic vessels
At stage CS17, PECAM$^+$/Prox1$^+$/VEGFR3$^+$ LECs were identified within the mesentery of the midgut, although no luminal lymphatic vessels were detectable (Fig. 4K,L). By stage CS18, the LECs in the mesentery had started forming luminal structures (Fig. 4M,N). At stage CS21, although lymphatic vessels were evident in the mesentery, only PECAM$^+$ blood vessels could be observed in the lamina propria of the intestine (Fig. 4O,P). This was still the case at CS23 (Fig. 4Q,R). By GW9, enlarged lymphatic vessels were seen in the mesentery (Fig. 4S,T), and a small number of PECAM$^+$/Prox1$^+$/VEGFR3$^+$ LECs were present in the lamina propria of the intestine (Fig. 4U,V). In summary, starting from CS17, LECs could be identified in the mesentery, and their numbers gradually increased, eventually resulting in the formation of large-diameter lymphatic vessels in the mesentery (Fig. 4W,X). PDPN and LYVE1 expression were detected in the lymphatic vessels within the mesentery and intestinal wall at GW9 (Fig. EV4X''',X'''',Y''',Y'''').

### Lymphatic vessel development in the lower jaw
Similar to the findings for cardiac lymphatic vessels, in mouse embryos, roughly 70–80% of the lymphatic vessels in the lower jaw originated from the cardiopharyngeal mesoderm (Maruyama et al, 2022). In line with this, the developmental pattern was resemble that of the lymphatic vessels surrounding the aorta. At stage CS16, PECAM$^+$/Prox1$^+$/VEGFR3$^+$ lymphatic vessels formed a small number of luminal structures without LYVE1 or PDPN expression (Fig. EV4A–C'''). The majority of LECs were present either as isolated or a few linked. This pattern persists at least until CS22 (Fig. EV4A–Q). At GW9, tubular lymphatic vessels are present in the subcutaneous tissue, expressing LYVE1 and PDPN (Fig. EV4X''''',Y''''').

### Kidney lymphatic vessel development
In mouse embryos, lymphatic vessels develop from the renal hilum within the metanephros, which matures into the adult kidney, and they gradually extend throughout the entirety of the kidney (Jafree et al, 2019). At stage CS23, we observed a few LECs in the renal hilum of metanephros. At GW9, increased lymphatic vessels were detected around blood vessels in the renal hilum with the expression of LYVE1 and PDPN (Fig. EV4R-U,4X'',Y'').

### Thoracic duct development
The thoracic duct is the largest lymphatic vessel in the human body, distributed along the posterior peritoneum, aorta, and esophagus.

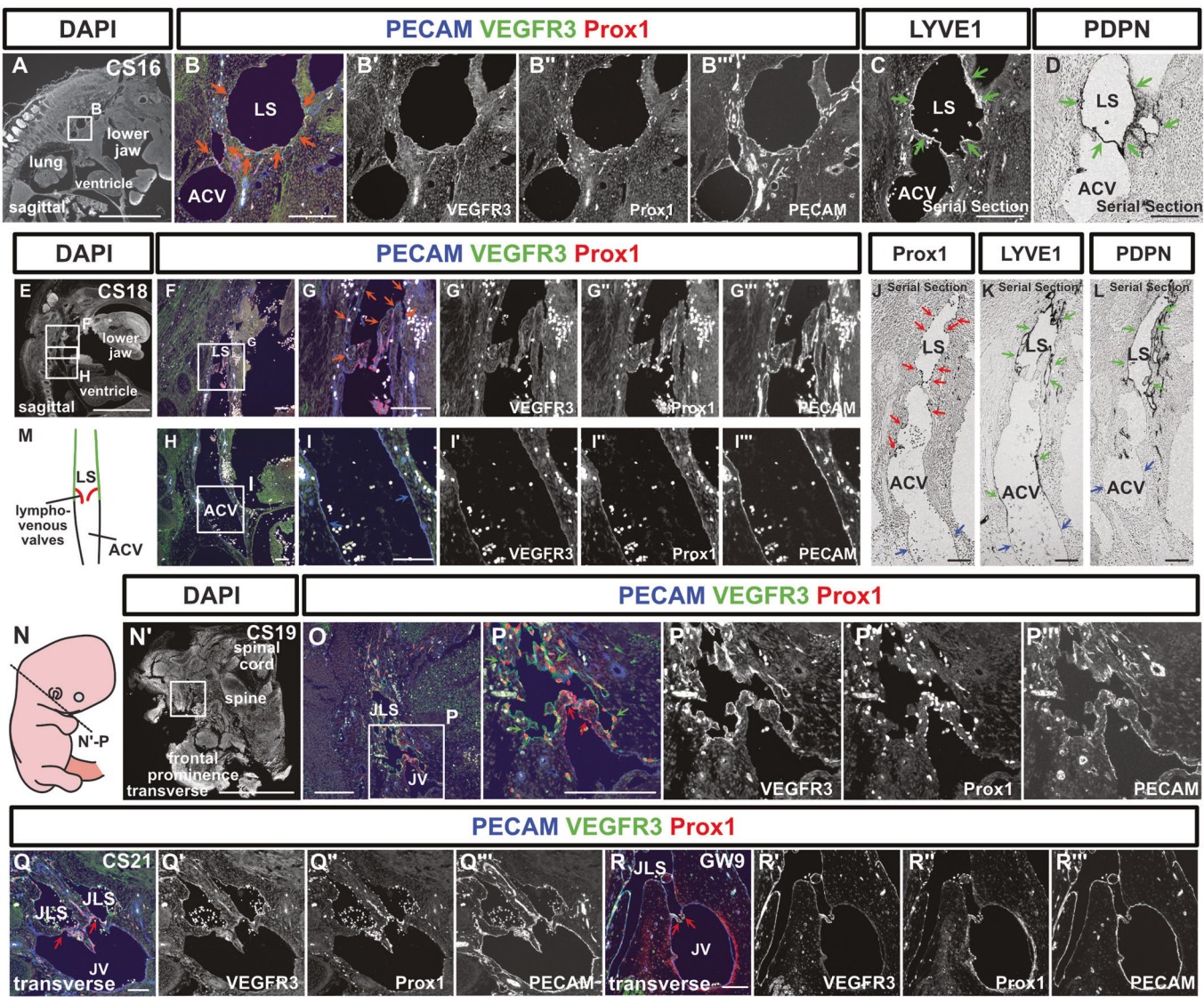

**Figure 2. Lymph sacs express LYVE1 and PDPN and establish connections with the anterior cardinal veins to form lymphovenous valves.**

(A–D) Immunostaining of sagittal sections from a CS16 embryo using the indicated antibodies. (B, C) Fluorescent immunostaining. (D) Immunostaining using the enzyme-antibody method. At CS16, lymph sacs emerged, which exhibited Prox1, VEGFR3, and PECAM expression (orange arrows). (C, D) The expression of LYVE1 and PDPN began within these lymph sacs at this stage (green arrows). (E–L) Immunostaining of sagittal sections from a CS18 embryo using the indicated antibodies. (E–I''') Fluorescent immunostaining. (J–L) The enzyme-antibody method. (E–I''') Lymph sacs expressing PECAM, VEGFR3, and Prox1 (orange arrows) formed connections with the ACVs, which expressed PECAM (blue arrows). (J–L) While Prox1 (red arrows), LYVE1, and PDPN (green arrows) were expressed in the lymph sacs, these markers were absent from the ACVs (blue arrows). There were four CS18-stage embryos, but only one included lymph sacs. (M) Schematic representation of the relationship between the ACVs and lymph sacs in a CS18 embryo. The lymph sacs and the ACVs were continuous, featuring a valve structure on the lymph sac side. (N–P''') Transverse sections and schematic representation of the positions of the sections in a CS19 embryo; Fluorescent immunostaining of PECAM, Prox1, and VEGFR3 was conducted. VEGFR3 was expressed in lymph sacs at CS19 (green arrows). The LVVs were overlaid with Prox1$^+$ cells (red arrows) (LVVs were identifiable in 1 of 5 embryos). (Q–R''') Transverse sections of a CS21 embryo and a GW9 fetus, highlighting the areas containing LVVs; as was the case at CS19, LVVs were visible (red arrows). (At CS21, LVVs were identified in 1 out of 2 embryos; at GW9, LVVs were identified in 1 out of 4 fetuses.) LS lymph sacs, ACV anterior cardinal vein, JLS jugular lymph sacs, JV jugular vein. Scale bars, 1 mm (A, E, N') or 100 µm (B–D, F–L, O–R). Source data are available online for this figure.

At GW9, large-diameter lymphatic vessels were observed around the aorta, which may eventually combine to form the future thoracic duct (Fig. EV4V,W).

### Proliferative activity of LECs during the organogenesis period

To examine LEC proliferation activity, we conducted Ki67, Prox1, and PECAM staining in specimens from CS13 and 16. At CS13, CV showed a Ki67 positivity in 19.6% of LECs, while extra-CV LECs exhibited 43.6%. At CS16, lymph sacs displayed a Ki67 positivity rate of 22.7%, and the LECs in the lower jaw were positive at a rate of 31% (Fig. EV5A–F''').

We display these comprehensive diagrams of early human lymphatic vessel development alongside comparative charts of mouse lymphangiogenesis as in Fig. 5.

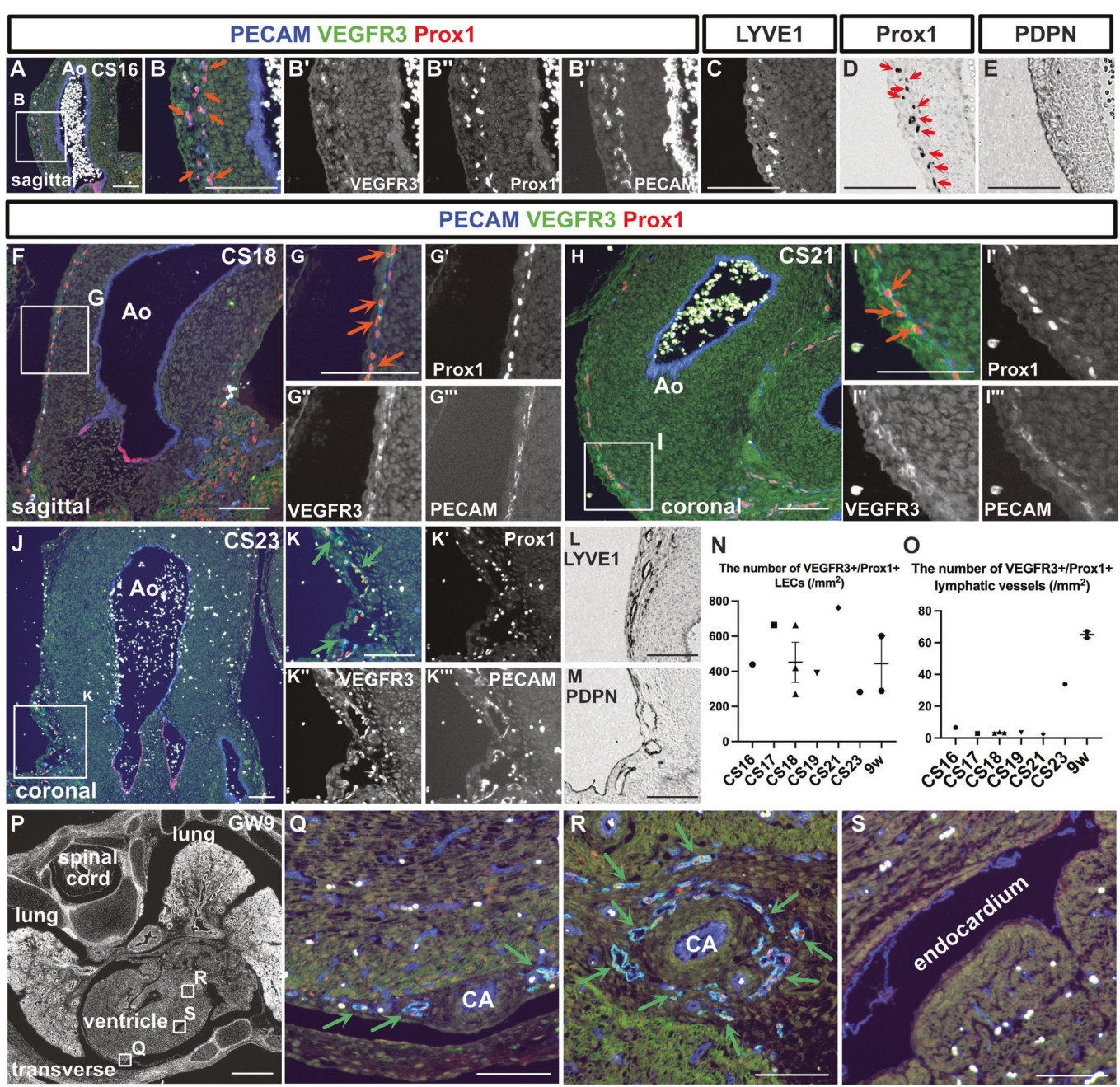

**Figure 3. Development of the cardiac lymphatic vessels.**

(A–E) Immunostaining of sagittal sections from a CS16 embryo using the indicated antibodies. (A–C) Fluorescent immunostaining. (D, E) Enzyme-antibody method, followed by DAB color development; PECAM$^+$/Prox1$^+$/VEGFR3$^+$ LECs were present in the aortic wall (orange arrows). (D) Prox1$^+$ cells were also identified by the enzyme-antibody method (red arrows). (F–I''') Immunostaining of sections from CS18 and CS21 embryos using the indicated antibodies. Consistent with the CS16 embryo, PECAM$^+$/Prox1$^+$/VEGFR3$^+$ LECs could be identified in the aortic wall (orange arrows). (J–M) Immunostaining of coronal sections from a CS23 embryo using the indicated antibodies. (J–K''') Fluorescent immunostaining. (L, M) Enzyme-antibody method, followed by DAB color development; LYVE1$^+$ and PDPN$^+$ lymphatic vessels were identified around the aorta (green arrows). (N, O) Changes in the number of PECAM$^+$/Prox1$^+$/VEGFR3$^+$ LECs in the aortic wall as development advanced (N). Changes in the number of PECAM$^+$/Prox1$^+$/VEGFR3$^+$ luminal lymphatic vessels as development advanced (O). (N, O) Data are presented as the mean ± standard error of the mean (SEM). Each dot represents a single individual. (P–S) Fluorescent immunostaining of PECAM, Prox1, and VEGFR3 in transverse sections of a GW9 fetus. By GW9, lymphatic vessels could be seen around the coronary arteries in the epicardium (green arrows). Ao aorta, CA coronary artery. Scale bars, 1 mm (P) or 100 µm (A–M, Q–S). Source data are available online for this figure.

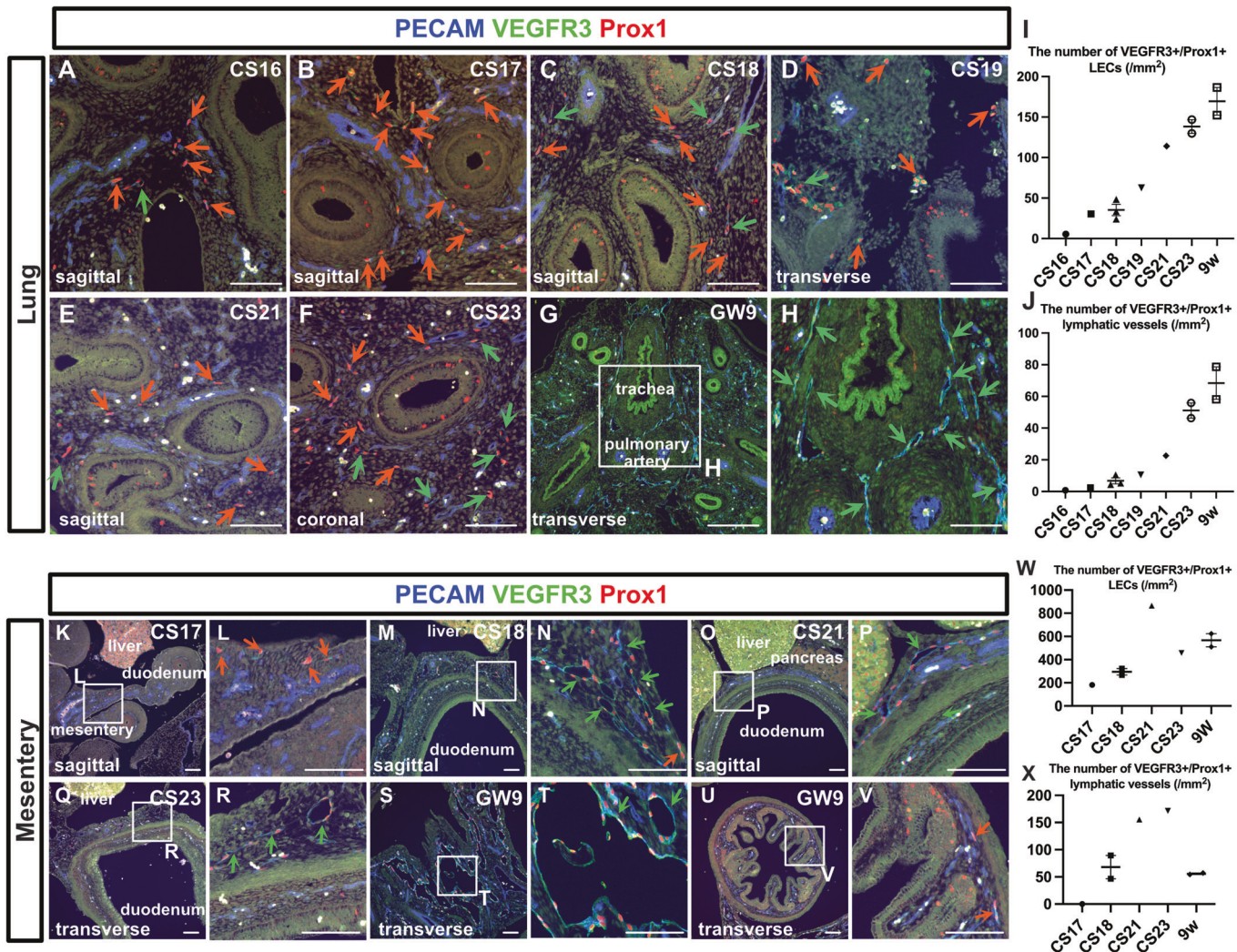

**Figure 4.   Development of the lung lymphatic vessels and mesenteric lymphatic vessels.**

(A–H) Immunostaining of PECAM, Prox1, and VEGFR3 in the lungs at each stage. The orange arrows indicate isolated or small clusters of LECs. The green arrows indicate lymphatic vessels that had formed luminal structures. (I, J) Changes in the number of PECAM$^+$/Prox1$^+$/VEGFR3$^+$ LECs in the aortic wall as development advanced (I). Changes in the number of PECAM$^+$/Prox1$^+$/VEGFR3$^+$ luminal lymphatic vessels as development advanced (J). Each dot represents a single individual. (I, J) Data are presented as the mean ± standard error of the mean (SEM). (K–V) Immunostaining of PECAM, Prox1, and VEGFR3 in the intestine and mesentery at each indicated stage. The orange arrows indicate isolated or small clusters of LECs that had not formed luminal structures, while the green arrows indicate lymphatic vessels that had formed luminal structures. (W, X) Changes in the number of PECAM$^+$/Prox1$^+$/VEGFR3$^+$ LECs in the aortic wall as development advanced (W). Changes in the number of PECAM$^+$/Prox1$^+$/VEGFR3$^+$ luminal lymphatic vessels as development advanced (X). (W, X) Data are presented as the mean ± standard error of the mean (SEM). Each dot represents a single individual. Scale bars, 100 μm (A–H, K–V). Source data are available online for this figure.

## Discussion

In this study, we examined the lymphatic vessel development process using human embryos. We identified antibodies that could be used in human embryos and fetuses. Using these antibodies, we identified endothelial cells expressing Prox1 and Coup-TF2, which are necessary for the production of early LECs, in the cardinal vein at CS12. On the other hand, while we tried to identify *Isl1*$^+$ LECs derived from the cardiopharyngeal mesoderm that we reported in mice (Maruyama et al, 2022), we were unable to find any LECs that simultaneously expressed both Isl1 and Prox1 at CS13, 14, and 16. This may have been due to the disappearance of Isl1 expression

before Prox1 expression. However, we observed Flk1$^+$/Isl1$^+$/PECAM$^-$ cells in the pharyngeal arch mesodermal region, and such cardiovascular progenitor cells (Milgrom-Hoffman et al, 2011) may form lymphatic vessels in the head and neck region by eventually expressing Prox1.

LECs budding from the cardinal vein gradually aligned and formed lymph sacs around the cardinal vein at CS13 to 16. LECs did not express LYVE1 and PDPN from CS12 to CS15; however, when they formed lymph sacs at CS16, they exhibited LYVE1 and PDPN expression, as well as Prox1 and VEGFR3. These marker expression patterns aligned with our previous works regarding to mice cardiac lymphatic vessel development, showing that LECs first

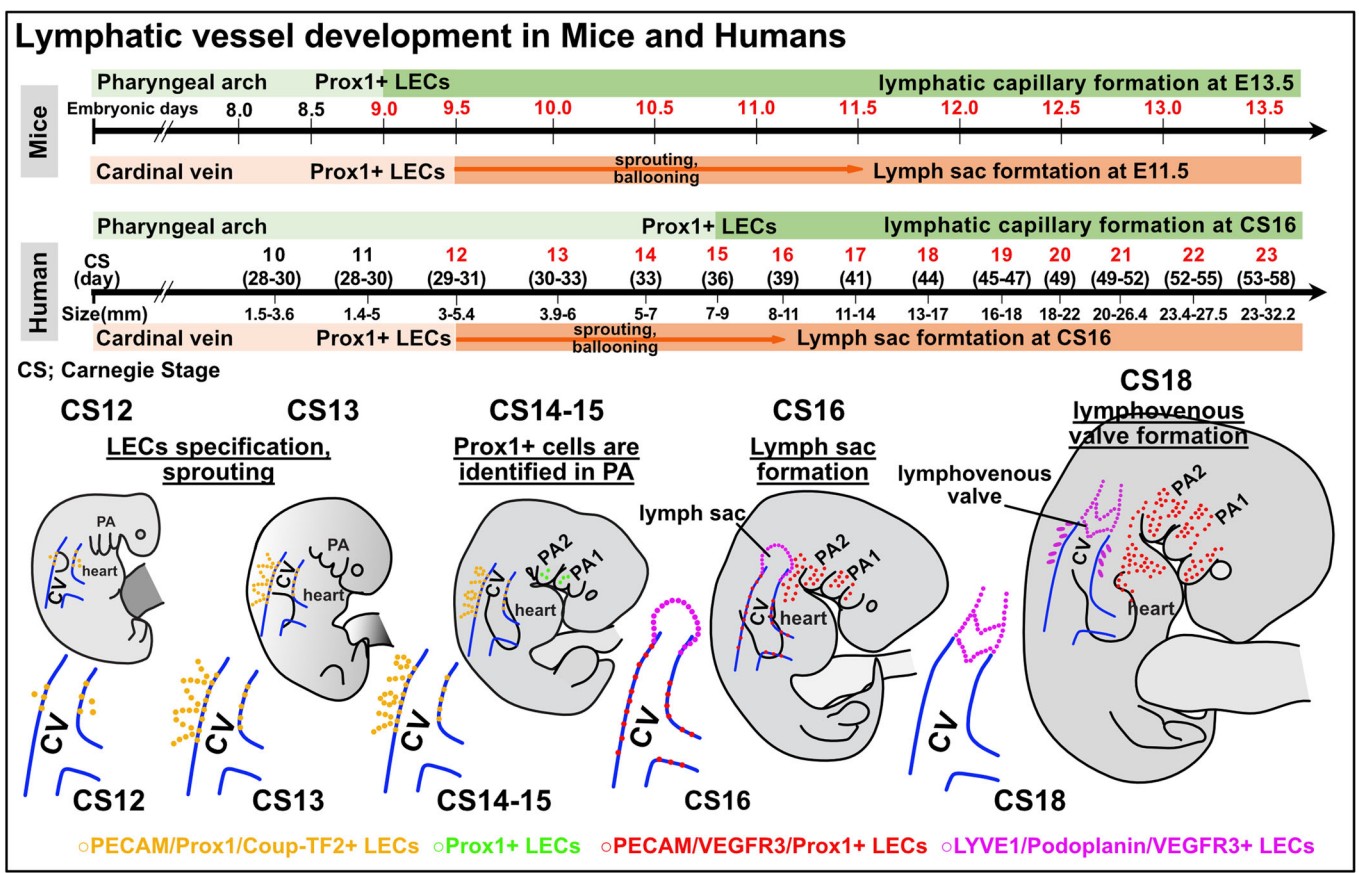

**Figure 5. Comparative overview of early lymphatic vessel development in mouse and human embryos.**

In mouse embryos, Prox1 expression is initiated in the cardiac pharyngeal mesoderm from E9.0 to E9.5, leading to the distribution of LECs in the head and neck, mediastinal, and cardiac outflow regions. By E13.5, these LECs start forming capillary lymphatics, which further develop into larger lymphatic vessels. Prox1 expression in the cardinal vein starts at E9.5, with the lymph sac forming post-sprouting by E11.5. In contrast, for human embryos, Prox1 expression in the pharyngeal arch area is first noted at CS15. Following this, LECs in the mandibular and cardiac outflow tract regions slowly establish a capillary lymphatic network, which by CS23 progresses to clearly defined lymphatic vessels with luminal structures. Prox1 expression in the cardinal vein begins at CS12, with the lymph sac being discernible by CS16. By CS18, the lymphovenous valve is identifiable between the lymph sac and the ACV. Across both embryos, early expression of transcription factors such as Prox1 and Coup-TF2 is observed, with the subsequent expression of VEGFR3, LYVE1, and Podoplanin. CS Carnegie stage, ACV the anterior cardinal veins.

express Prox1, then VEGFR3, followed by the expression of LYVE1 (Maruyama et al, 2019, 2021). We also confirmed that LVVs formed at the junction between the lymph sacs and the cardinal veins at CS18. LVVs initially showed an irregular shape, but gradually formed smooth bicuspid valves around GW9. The proper formation of LVVs is critical as a cause of primary lymphedema in humans (Tammela and Alitalo, 2010). Therefore, the understanding of the normal LVVs development from this study may help to elucidate the pathology of primary lymphedema caused by genetic mutations (Oliver et al, 2020).

Focusing on the lymphatic vessel development of each organ, for example, the lymphatic vessels in the lower jaw and cardiac outflow tract, which we reported to be derived from the cardiopharyngeal mesoderm, did not exhibit noticeable luminal formation at CS16-21. At CS22, many LECs still capillary lymphatics. In the heart, LECs gradually gathered together, luminal formation accelerated, and PDPN and LYVE1 expression were observed at CS23. In the lower jaw, the expression of PDPN and LYVE1 and the formation of luminal structures were confirmed at GW9. In the lungs, the

number of LECs increased and tubular structures started to form as development progressed. By GW9, we were able to confirm the presence of lymphatics around the bronchi, in a similar distribution to that seen in adults. At CS17, isolated LECs were observed within the mesentery, which was continuous with the posterior abdominal wall. At this stage, although lymph sacs had developed within the posterior abdominal wall, no continuity with the isolated LECs was observed. As development progressed, the mesenteric LECs formed large lymphatic vessels. However, the presence of LECs in the intestinal wall was not observed until GW9. In the kidney, considering that only a few LECs were observed around CS23, the lymphatics may begin to develop around CS23 in human embryos. Also, in order to consider the development of the thoracic duct, we confirmed the presence of large-diameter lymphatic vessels around the aorta and esophagus at GW9. However, the course and shape of the thoracic duct vary greatly among individuals, so a more detailed analysis will likely be needed to determine how it develops. Given that genetic cell lineage analysis in humans requires significant challenges, combining

developmental analysis with multi-omic analysis, such as single-cell RNA-seq analysis, could potentially clarify the details of organ-specific lymphatic vessel development.

Although recent research has shown that the meningeal lymphatic vessels play a crucial role in the pathophysiology of the central nervous system (CNS) (Oliver et al, 2020), we were unable to confirm the presence of the lymphatic vessels in the brain and spinal cord. Indeed, in mice the CNS lymphatic vessels develop after birth (Antila et al, 2017).

The regulation of Prox1 expression requires transcription factors, such as Coup-TF2 (Yamazaki et al, 2009) and Sox18 (François et al, 2008) to the promoter region. We confirmed the expression of Coup-TF2 in the cardinal vein and surrounding LECs; however, the expression of Sox18 in the cardinal vein and LECs was not observed in human embryos at CS13. This might be due to the expression timing of Sox18 is more restricted in the cardinal vein in humans. In the process of LVVs formation, in addition to Prox1, the upregulation of FOXC2 and GATA2 leads to the acquisition of the differentiation fate towards LVV endothelial cells (Geng et al, 2016). We attempted to elucidate the expression patterns of these molecules, but the number of sections containing LVVs in a single individual was limited; therefore, we were not able to analyze these expression patterns.

In summary, this study clarified the process underlying early lymphatic vessel formation in human embryos. LECs originate from embryonic veins and form lymph sacs, a process which has been clarified in mice and zebrafish and is now confirmed to also occur in humans. On the other hand, the lymphatic vessel development processes for each organ vary in terms of speed and marker expression, possibly due to differences in cellular origin and signaling. This study is important for elucidating the evolutionarily conserved processes of lymphatic vessel formation and for aiding the extrapolation of findings from animals to humans.

## Methods

### Tissue collection and ethical considerations

For this study, 31 preserved human embryos in organogenesis period and 3 fetuses ranging from CS8 to GW9 were analyzed. The samples were collected for pathological examination, which entails histological verification of pregnancy (investigating the decidua's interstitial and the endometrium's Arias-Stella reactions) and confirmation of fetal components (examination of chorionic tissue). In addition, they were assessed to ensure the absence of conditions such as hydatidiform mole or choriocarcinoma. We selected residual specimens, collected between January 2000 and October 2021, that included embryos and utilized these specimens specifically for the purposes of this research. The use of human samples was approved by the Ethics Committee of Mie University Hospital (approval number: H2021-228). Informed consent was acquired at the time of surgery, complemented by the availability of an opt-out provision, ensuring participants' autonomy to withdraw from the study at any point. The staging of the embryos and fetuses used in the experiments was done using a combination of the

**Table 1. Information of human embryos and fetuses.**

| | Number and characteristics of embryos and fetuses. | Cause and additional information of embryos and fetuses |
|---|---|---|
| CS9-10 | $N = 1$. The heart tube formation. | Miscarriage. Five weeks from LMP. |
| CS10 | $N = 2$. The neural tube formation. | Miscarriage ($N = 2$). Details unknown. The patients bring miscarriage specimens. |
| CS11 | $N = 1$. GL is 1.4–5 mm. Precardinal veins formation. | Tubal pregnancy. GL was 5 mm (transvaginal ultrasound). Five weeks from LMP. |
| CS12 | $N = 1$. GL is 3–5.4 mm. Trabecular formation in the heart. | The samples were taken by emergency surgery due to tubal pregnancy. |
| CS13 | $N = 2$. GL is 3.9–6 mm. The lung buds formation. | The samples were taken by emergency surgery due to tubal pregnancy. |
| CS14 | $N = 2$. GL is 5–7 mm. | Miscarriage ($N = 2$). Six weeks from LMP. GL was 6 mm ($N = 1$, transvaginal ultrasound). The size of the other was unknown. |
| CS15 | $N = 1$. GL is 7–9 mm. The lens pit formation. | Miscarriage. Seven weeks from LMP. GL was 5.5 mm (transvaginal ultrasound). |
| CS16 | $N = 1$. GL is 8–11 mm. The retinal pigment is distinct. | The samples was taken by emergency surgery due to tubal pregnancy. |
| CS17 | $N = 1$. GL is 11–14 mm. | Tubal pregnancy. Eight weeks from LMP. GL was 14.0 mm (transvaginal ultrasound). |
| CS18 | $N = 4$. GL is 13–17 mm. The posterior cardinal vein has disappeared. | Artificial abortion ($N = 3$), tubal pregnancy ($N = 1$). Eight weeks from LMP. GL was 14.0, 15.0, and 16.5 mm (transvaginal ultrasound). The other was unknown. |
| CS19 | $N = 5$. Embryos have a greatest length of 16–18 mm. | Artificial abortion ($N = 5$), tubal pregnancy ($N = 1$). Eight weeks from LMP. GL was 17 mm ($N = 1$, the actual size of the embryo). The others were unknown. |
| CS20 | $N = 1$. The CRL is around 17.0–19.3 mm. | Artificial abortion. Eight weeks from LMP. CRL was 18.7 mm (transvaginal ultrasound) |
| CS21 | $N = 2$. The CRL is around 18.8–23.0 mm. | Miscarriage and tubal pregnancy. Nine weeks from LMP. CRL = 23 mm ($N = 1$, transvaginal ultrasound). The other was unknown. |
| CS22 | $N = 3$. The CRL is around 22.0–24.0 mm. A few large glomeruli are present in the kidney. | Artificial abortion ($N = 3$). Ten weeks from LMP. No ultrasound information was available. |
| CS23 | $N = 4$. The CRL is around 23.5–28.0 mm. Most organs are fully formed. | Artificial abortion ($N = 4$). In all, 10–11 weeks from LMP. CRL = 29.4 mm ($N = 1$). The others were unknown. |
| GW9 | $N = 3$. The CRL is around 25–45 mm (Papageorghiou et al, 2014). | Artificial abortion ($N = 1$), miscarriage ($N = 2$). CRL was 40.0 and 39.4 mm (transvaginal ultrasound). The actual size was 42 mm ($N = 1$). |

*GL* greatest length of the embryo, *CRL* the crown-rump length, *LMP* the last menstrual period.

Carnegie stage (CS) (O'Rahilly and Müller, 2010) and clinical information: (1) menstrual weeks, (2) morphology, (3) length (crown-rump), and (4) anatomical features (we also referred to the detailed anatomical information on this website: https://www.ehd.org/virtual-human-embryo/). Detailed information regarding each sample is presented in Table 1. The sex of each sample was not determined, with the exception of one case of miscarriage. In this particular case, chromosomal analysis verified the absence of any karyotypic abnormalities. There were no malformations observed in any of the embryos or fetuses. Nevertheless, for the remaining embryos, there is a possibility that developmental defects or mutations could lead to abnormalities in the lymphatic vessels.

### Section preparation and screening for lymphatic endothelial cells

The embryos and fetuses were fixed with 10% formalin neutral buffer solution at 4 °C for 1 to 2 days. The specimens embedded in paraffin were cut into 1-µm thick sections until the sample was exhausted. Hematoxylin and eosin (HE)-stained sections were created at 10-µm intervals to confirm the anatomical structures of the specimens. Considering the impact of autofluorescence, all staining procedures were coupled with enzyme immunohistochemistry (IHC) to confirm the specificity of the signals. To screen for lymphatic vessels, immunostaining of Prox1, CD31, and LYVE1 was performed on one out of every ten sections using the enzyme IHC. After stage CS16, immunostaining of PDPN was also performed when screening for lymphatic vessels.

### Immunohistochemistry (IHC)

HE staining and IHC were performed using 1-µm thick sections. In IHC, sections were deparaffinized and rehydrated through a series of xylene and ethanol. For the enzyme-antibody method, endogenous peroxidase activity was blocked using 0.3% hydrogen peroxide ($H_2O_2$) in methanol for 20 min. In fluorescent antibody staining, to suppress autofluorescence, samples were incubated in 0.1% sodium borohydride in 0.1 M phosphate-buffered saline (PBS) (137 mM NaCl, 2.7 mM KCl, 10 mM $Na_2HPO_4$, and 1.8 mM $KH_2PO_4$, pH = 7.2) for 30 min, then rinsed with water, and subsequently incubated for 5 min in 0.2 M glycine in 0.1 M PBS. Antigen retrieval was carried out using a pressure chamber with Tris-EDTA buffer (7.4 mM Tris, 1 mM EDTA-2Na, pH 9.0). Slides were incubated with primary antibodies against Prox1 (11-002, AngioBio, 1:150, RRID:AB_10013720), Prox1 (AF2727, R&D Systems, 1:150, RRID:AB_2170716), LYVE1 (ab14917, abcam, 1:150, RRID:AB_301509), VEGFR3 (AF349, R&D Systems, 1:100, RRID:AB_355314), Coup-TF2 (EPR18443, abcam, 1:150, RRID:AB_2895604), D2-40 (anti-PDPN) (413151, nichirei biosciences, at its original concentration (ready to use product)), Flk1 (AF357, R&D Systems, 1:150, RRID:AB_355320), Isl1 (PA5-27789, Invitrogen, 1:200, RRID:AB_2545265), Ki67 (ab15580, abcam, 1:500, RRID:AB_443209) and PECAM (M0823, DAKO, 1:100, RRID:AB_2114471). In the enzyme-antibody method, the secondary antibody from the Histofine Simple Stain System (Nichirei Biosciences) was incubated with the slides for 1 h. Peroxidase activity was visualized using DAB-$H_2O_2$. For fluorescent immunostaining, Alexa Fluor-conjugated secondary antibodies (Abcam, 1:400) were subsequently applied. After this, True Black (Biotium) was used according to the manufacturer's instructions to suppress autofluorescence. Imaging was carried out using a Keyence BZ-X700 microscope. All images were processed using ImageJ software.

### Statistical analysis

The numbers of LECs and lymphatic vessels are represented as the average of two or more immunostained slides. Image processing was conducted using ImageJ (NIH). Data are presented as the mean ± standard error of the mean (SEM).

## Data availability

This study includes no data deposited in external repositories.

## Peer review information

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

## Acknowledgements

The authors thank all of the laboratory members for their helpful discussion and encouragement. We also thank Dr. Hiroki Kurihara (the University of Tokyo) for discussing the contents of the paper. This study was supported in part by Grants-in-Aid for Scientific Research from the Ministry of Education, Culture, Sports, Science, and Technology, Japan (20K17072 and 23K15949 to KM); the Japan Foundation for Applied Enzymology (VBIC to KM); the SENSHIN Medical Research Foundation (KM); Mochida Memorial Foundation for Medical and Pharmaceutical Research (KM); Japan Agency for Medical Research and Development (AMED) under Grant Number 22jm0610079h0001 (KM); Takeda Medical Research Foundation (KM); Mie University: Research promotion and graduate school reform-related research grant project <Interdisciplinary Collaborative Research Support Project> (KM); TERUMO life science foundation (KM); Kurozumi medical foundation (KM).

## Author contributions

**Shoichiro Yamaguchi**: Conceptualization; Investigation. **Natsuki Minamide**: Investigation. **Hiroshi Imai**: Resources. **Tomoaki Ikeda**: Resources. **Masatoshi Watanabe**: Resources. **Kyoko Imanaka-Yoshida**: Supervision. **Kazuaki Maruyama**: Conceptualization; Data curation; Formal analysis; Supervision; Funding acquisition; Investigation; Writing—original draft; Project administration; Writing—review and editing.

## Disclosure and competing interests statement

The authors declare no competing interests.

# Expanded View Figures

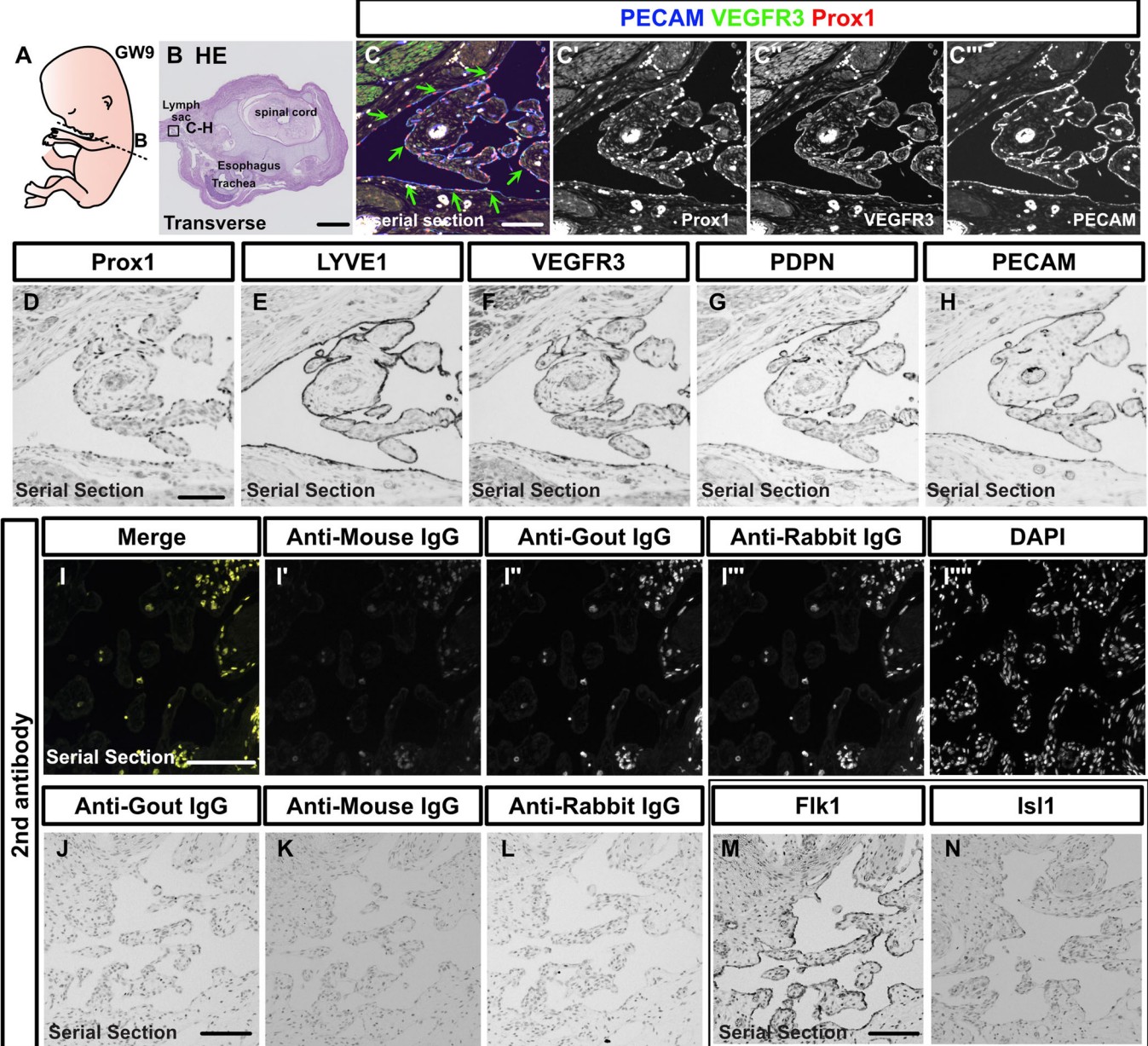

**Figure EV1. Multiple lymphatic markers are expressed in fetal lymph sacs.**

(A–H) Schema showing the positions of the sections in a GW9 fetus (**A**), and immunostaining of transverse sections (**B–H**); (C–C‴) Fluorescent immunostaining of PECAM, Prox1, and VEGFR3; These markers were expressed in lymph sacs (green arrows). (D–H) Immunostaining of Prox1, LYVE1, VEGFR3, PDPN, and PECAM using the enzyme-antibody method, with color development by DAB. (I–L) Imaging of the secondary antibody-only staining. (M, N) Immunostaining of Flk1, and Isl1 using the enzyme-antibody method, with color development by DAB. Scale bars, 1 mm (**B**) or 100 μm (**C, D, I, J, M**).

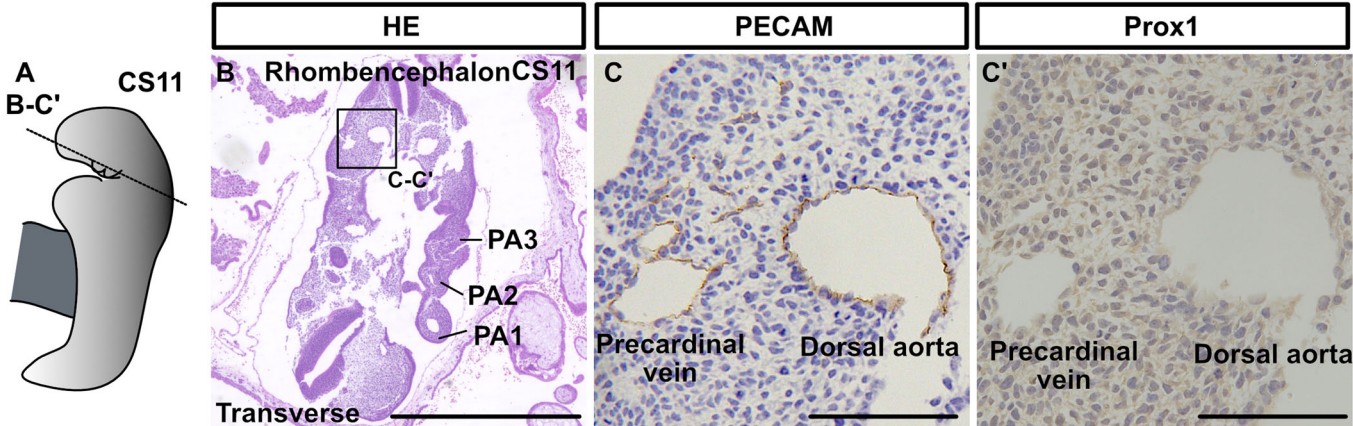

**Figure EV2.** Prox1 expression is not observed in the precardial vein of the CS11 embryo.

(**A–C'**) Immunostaining of transverse sections of a CS11 embryo with the indicated antibodies and schema showing a CS11 embryo ($n = 1$). Scale bars, 1 mm (**B**) or 100 μm (**C, C'**).

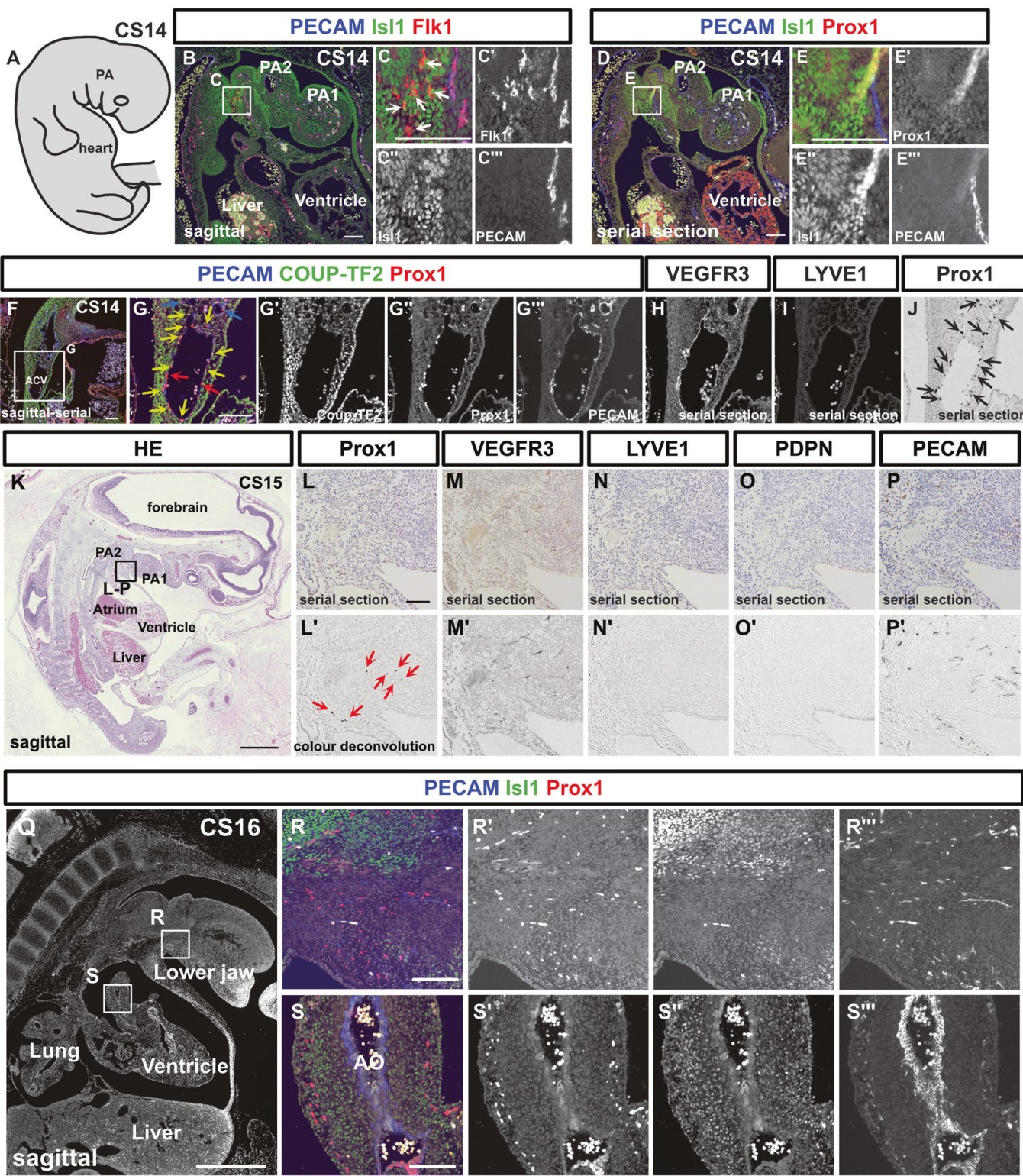

◀ **Figure EV3.** **LECs bud from the cardinal veins and form luminal structures.**

(**A–J**) Immunostaining of sagittal sections of a CS14 embryo with the indicated antibodies and schema showing the CS14 embryo. (**B–C‴**) Cardiovascular progenitor cells, which were composed of FLK1$^+$/Isl1$^+$/PECAM$^-$ cells, were observed in the second pharyngeal arch (white arrows). (**F–G‴**) PECAM$^+$/Prox1$^+$/Coup-TF2$^+$ cells (yellow arrows: Frequency of Coup-TF2$^+$ cells among PECAM$^+$/Prox1$^+$ cells=44.1%, $n = 1$ [the ACVs could not be identified in another embryo]), and PECAM$^+$/Prox1$^+$/Coup-TF2$^-$ cells (red arrows) were observed in and around the ACVs. (**K–P′**) HE staining of a sagittal section of a CS15 embryo (**K**) and immunostaining of sagittal sections of the same CS15 embryo with the indicated antibodies (**L–P′**); At this stage, scattered Prox1$^+$ cells were observed in the pharyngeal arch (red arrows). (**Q–S‴**) Immunostaining of sagittal sections of a CS16 embryo with the indicated antibodies. PA1 first pharyngeal arch, PA2 second pharyngeal arch, ACV anterior cardinal vein. scale bars, 1 mm (**K**) or 100 μm (**B–G**, **L**).

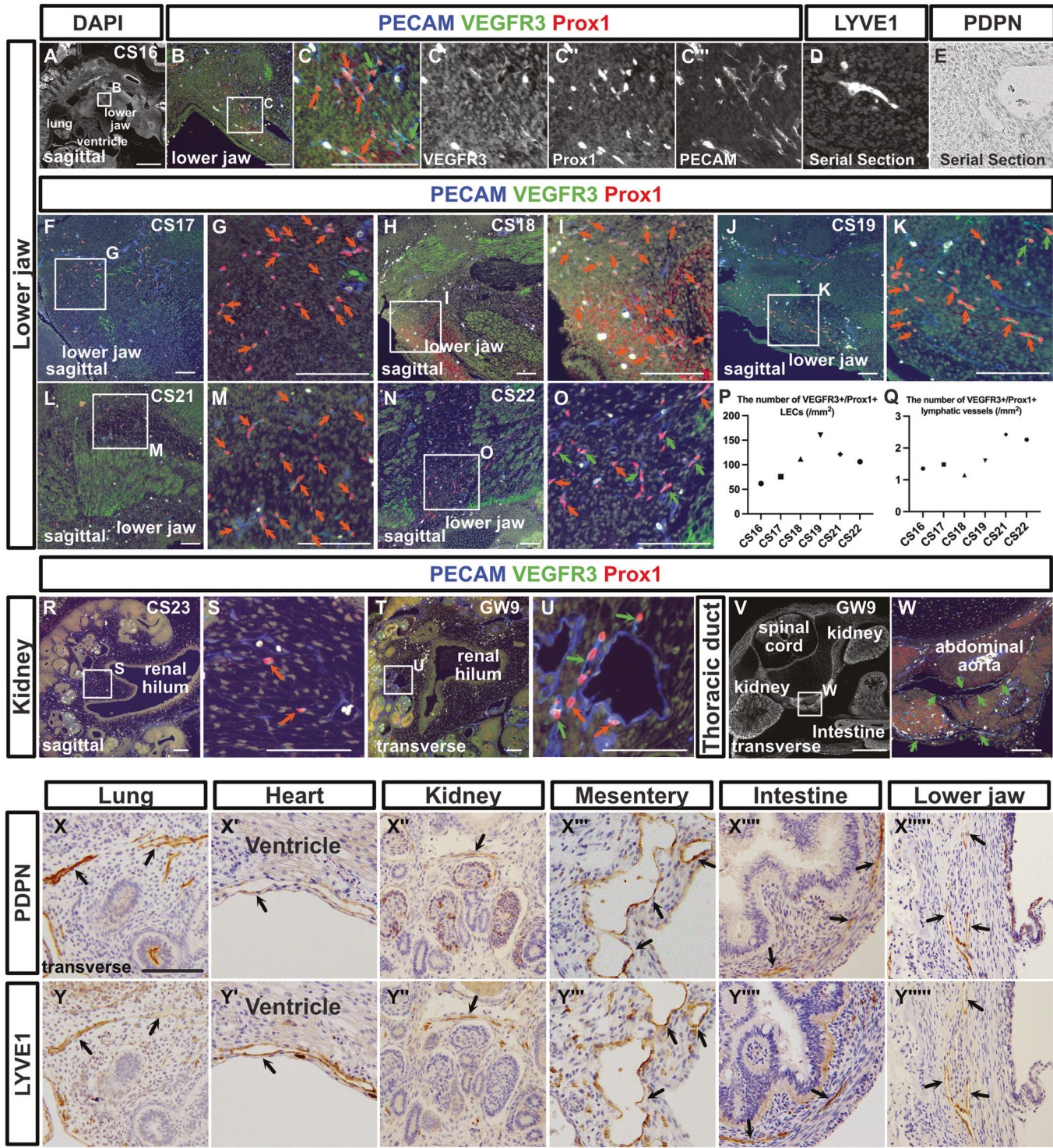

**Figure EV4. Development of lymphatic vessels in the lower jaw, kidneys, and thoracic duct.**

(A–O) Immunostaining of sagittal sections of embryos collected at the described stages with the indicated antibodies; The orange arrows indicate isolated or small clusters of LECs. The green arrows indicate lymphatic vessels that had formed luminal structures. (P, Q) Changes in the number of PECAM$^+$/Prox1$^+$/VEGFR3$^+$ LECs in the aortic wall (P). Changes in the number of PECAM$^+$/Prox1$^+$/VEGFR3$^+$ luminal lymphatic vessels (Q). Each dot represents a single individual. (R–U) Fluorescent immunostaining of PECAM, Prox1, and VEGFR3 in sections from embryos collected at the described stages. The orange arrows indicate isolated or small clusters of LECs. The green arrows indicate lymphatic vessels that had formed luminal structures. (V, W) The green arrows indicate lymphatic vessels surrounding the aorta at GW9. (X, Y) Immunostaining for PDPN and LYVE1 in the lung, heart, kidney, mesentery, intestine, and lower jaw. Scale bars, 1 mm (A, V) or 100 µm (B, C, F–O, R–U, W, X).

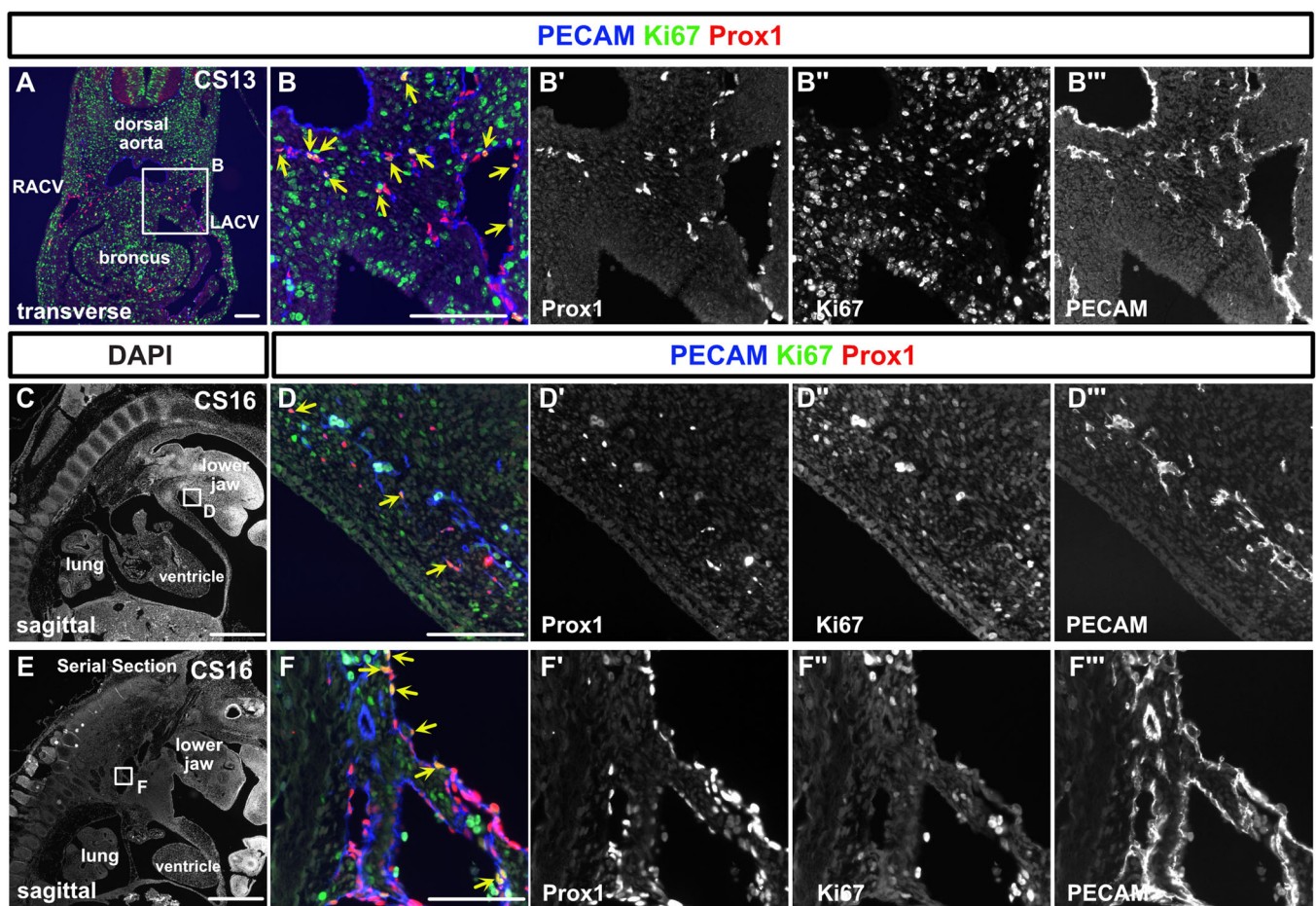

**Figure EV5. Proliferation activity of lymphatic endothelial cells at CS13 and CS16.**

(A–F''') Immunostaining of transverse and sagittal sections using the indicated antibodies in a CS13 and 16 embryos. We detected Prox1$^+$/PECAM$^+$/Ki67$^+$ LECs (indicated by yellow arrows) as well as Prox1$^+$/PECAM$^+$/Ki67$^-$ LECs. At CS13, within CV, 19.6% of LECs were Ki67 positive, while outside the CV, the percentage was 43.6%. At CS16, 22.7% of LECs in the lymph sacs were Ki67 positive, and in the lower jaw, the positivity rate was 31% (based on an average of two sections at $n = 1$). CV cardinal vein, RACV right anterior cardinal vein, LACV left anterior cardinal vein. scale bar, 1 mm (**C**, **E**), 100 μm (**A**, **B**, **D**, **F**).

