## [Peer Review File · The EMBO Journal]

The development of early human lymphatic vessels as characterized by lymphatic endothelial markers

Shoichiro Yamaguchi, Natsuki Minamide, Hiroshi Imai, Tomoaki Ikeda, Masatoshi Watanabe, Kyoko Imanaka-Yoshida, and Kazuaki Maruyama

DOI: [10.15252/embj.2023116122](https://doi.org/10.15252/embj.2023116122)

Corresponding author: [Kazuaki Maruyama \(k-maruyama0608@med.mie-u.ac.jp\)](mailto:k-maruyama0608@med.mie-u.ac.jp)

Review Timeline:

Transfer from Review Commons:	9th Nov 23
Editorial Decision:	24th Dec 23
Revision Received:	2nd Jan 24
Accepted:	8th Jan 24

Editor: *Kelly Anderson*

Transaction Report:

Review
COMMONS

This manuscript was transferred to The EMBO Journal following peer review at Review Commons.

Review #1

1. Evidence, reproducibility and clarity:

Evidence, reproducibility and clarity (Required)

Summary

Yamaguchi et al. performed a comprehensive characterisation of lymphatic vessel development in human embryos, spanning stages C8 to GW9. Through the utilisation of immunohistochemistry targeting proteins expressed in the lymphatic endothelium and blood endothelium, the authors have discerned the presence of lymphatic endothelial cells within the cardinal vein and in extravascular locations. By systematically analysing the progression of embryonic stages, the authors identified the emergence of lymph sacs. Furthermore, they confirmed the presence of lymphatics in various organs, such as the heart, kidney, lung, and mesentery. However, lymphatics were not detected in the central nervous system during the embryonic stages. At the molecular level, lymphatic endothelial cells express similar factors as in mice, including Prox1, Vegfr3, Lyve1, and PDPN, although the timing and combination of these factors may vary depending on the tissue. This study significantly contributes to our knowledge of lymphatic development in humans.

Major comments

Human embryo samples are exceptionally valuable and ethically sensitive, making their maximum utilisation crucial. While the authors conducted a thorough anatomical and molecular analysis, it raises questions about whether more insights can be gleaned.

Specifically, the authors should clarify whether data from embryos collected at CS8-CS10 were processed, and what was the status of venous and lymphatic development?

The authors commented that CS11 lymphatic vessels were not identified in the vein. Was there any indication of LECs outside the vein? Could the authors include images of this stage?

For embryos at CS12, it would be insightful to know the proportion of LECs versus VECs within the vein, the quantity of LECs outside the veins, and whether there was section-dependent variability in these observations.

It would be helpful if Table 1, "Information of human embryos and fetuses", could be complemented with a summary of the main findings at each stage, including which markers LECs expressed and their distribution.

The authors mentioned differences in lymphatic markers at various regions of the embryo and different developmental stages. It is essential to clarify whether all regions express the same markers at the latest developmental stage.

To strengthen the assertion that this study provides unique insights compared to those of mice, a schematic summarising the similarities and differences between mouse and human observations should be included.

A discussion of the limitations of analysing embryos from abnormal pregnancies is necessary. In addition to the determined lack of chromosomal abnormalities, it is crucial to consider phenotypical and morphological integrity. The authors should address the possibility of developmental defects and mutations causing abnormalities in the lymphatic vessels.

****Minor comments****

In the abstract, the authors refer to lymphatic malformations as a specific type of lymphatic disease. We recommend acknowledging the broader implications of this study beyond such specific cases.

The term "lymph-related disease" should be clarified for better understanding.

Figure 3S shows kidney samples, not the myocardium or endocardium, as indicated.

2. Significance:

Significance (Required)

This study largely reaffirms the existing knowledge from mouse models and previous human data. Given the absence of a cure for lymphatic diseases, gaining a deeper understanding of how lymphatic vessels develop in humans could serve as a crucial stepping stone in this field of research.

3. How much time do you estimate the authors will need to complete the suggested revisions:

Estimated time to Complete Revisions (Required)

(Decision Recommendation)

Between 1 and 3 months

4. *Review Commons* values the work of reviewers and encourages them to get credit for their work. Select 'Yes' below to register your reviewing activity at Web of Science

Reviewer Recognition Service (formerly Publons); note that the content of your review will not be visible on Web of Science.

No

Review #2

1. Evidence, reproducibility and clarity:

Evidence, reproducibility and clarity (Required)

This study by Yamaguchi et al., explores the progression of lymphatic vessel growth in different stages of human embryos. They also try to identify the origin of the lymphatic vessels in different organs. The study first shows that lymphatic endothelial cells (LECs) first show up in the anterior cardinal veins (ACVs) of CS12 in human embryos, which is similar to what is known to occur in mouse embryos. They also checked whether the PROX1+ LECs of the heart are derived from Flk1+/Isl+/PECAM- cells. However, Flk1+/Isl+/PECAM- cells do not co-express PROX1. These results suggest that in human embryos LECs originate from the ACVs. The authors then identify that lympho-venous valves formed between lymph sacs and the cardinal veins at around Carnegie Stage (CS)18. The valves have showed obvious bicuspid shape at Gestational week (GW)9. Finally, the authors demonstrate that the development of lymphatic vessels happens at different time points in various organs. At CS16, lymphatic vessels and LECs can be detected in the lower jaw, heart and the lungs; mesenteric and intestinal lymphatic vessels can be detected between CS17 and CS18; kidney lymphatic vessels can be found at CS23; At GW9, the lymphatic vessels are observed around the aorta, which may combine to form the future thoracic duct. Together, this informative study sheds light on the progression of lymphatic vasculatures during embryonic stage in humans.

This study has many strengths, in addition to some areas that if addressed, would further increase the impact of the findings. These include:

1. Since immunostaining is the major method that the authors have used for their work, they could use positive and negative controls (secondary antibody only or IgG control) for different antibodies. The authors can also show some Isl1 and Flk1 staining in GW9 fetus or adult tissue, like PROX1 or LYVE1 in Supplemental figure 1.
2. Figures 1 F-H, S' and S'', U', U'', and U''' are hard to appreciate. Can the authors offer higher quality images or show some confocal images?
3. According to the author's previous publications (ref 17 and 30) and literature (ref 31), Flk+/Isl+/PECAM- cells differentiate into LECs. However, in this work they did not observe any PROX1+Isl1+ cells at CS13 and CS14. I am curious to know if they found any PROX1+Isl1+ cells at later time points such as GW9.
4. Figure 3 N and O show comparable VEGFR3+PROX1+ cell numbers in different time

points, however it shows increased VEGFR3+PROX1+ vessel numbers. If so, do LECs become more elongated and form the vessel-like structures?

5. The authors have mentioned that the staging of the embryos and fetuses was done by Carnegie stage and clinical information. The authors should offer more detailed information about those embryos and fetuses. For example, crown-rump length, menstrual weeks, craniofacial features etc. This information will be useful for other researchers in this field.

2. Significance:

Significance (Required)

Strengths: Very informative results for human embryonic lymphatic development. They have performed the experiments at various developmental stages.

Limitations: Image quality need to be improved. Many high magnification images are not clear. Human samples come from certain diseases, which might have affected the embryo's development.

Advance: this study clarified the process of early lymphatic vessel formation in human embryos.

Audience: clinical and basic science in developmental biology and lymphatic biology.

Reviewer expertise: lymphatic development, lymphatic biology, vascular biology.

3. How much time do you estimate the authors will need to complete the suggested revisions:

Estimated time to Complete Revisions (Required)

(Decision Recommendation)

Between 1 and 3 months

Yes

Revision Plan

Manuscript number: #RC-2023-02149

Corresponding author(s): Kazuaki Maruyama

1. General Statements [optional]

Dear Reviewers #1 and #2,

We extend our deepest gratitude for your dedication to reviewing our manuscript during such a busy period. We have diligently addressed the insightful feedback provided in our revisions. The variable quality of human fetal tissues, due to fixation and extended preservation times, is acknowledged as a limitation that may affect the quality of our immunostaining results. Despite this, we maintain that the findings from these experiments are crucial for human applications. The extrapolation of the results from mice experiments to human biology is a critical step in propelling research forward. We are confident that our paper, with its acknowledged limitations, still offers valuable contributions to our understanding in this domain.

Please find the primary amendments of our revision detailed below for your review.

2. Description of the planned revisions

Since most of the revisions are complete, I will include them in section 3.

3. Description of the revisions that have already been incorporated in the transferred manuscript

Reviewer #1 (Evidence, reproducibility and clarity (Required)):

Summary

Yamaguchi et al. performed a comprehensive characterisation of lymphatic vessel development in human embryos, spanning stages C8 to GW9. Through the utilisation of immunohistochemistry targeting proteins expressed in the lymphatic endothelium and blood endothelium, the authors have discerned the presence of lymphatic endothelial cells within the cardinal vein and in extravascular locations. By systematically analysing the progression of embryonic stages, the authors identified the emergence of lymph sacs. Furthermore, they confirmed the presence of lymphatics in various organs, such as the heart, kidney, lung, and mesentery. However, lymphatics were not detected in the central nervous system during the embryonic stages. At the molecular level, lymphatic endothelial cells express similar factors as in mice, including Prox1, Vegfr3, Lyve1, and PDPN, although the timing and combination of these factors may vary depending on the tissue. This study significantly contributes to our knowledge of lymphatic

development in humans.

Major comments

Human embryo samples are exceptionally valuable and ethically sensitive, making their maximum utilisation crucial. While the authors conducted a thorough anatomical and molecular analysis, it raises questions about whether more insights can be gleaned.

Specifically, the authors should clarify whether data from embryos collected at CS8-CS10 were processed, and what was the status of venous and lymphatic development?

Response:

After a careful review of the clinical data for the specimen previously classified as CS8, we found a record indicating the initial detection of a heartbeat in the preceding week, an observation not made earlier. When correlating the last menstrual period with the morphological features, such as the open neural tube, it suggests that the specimen may actually be at CS 9-10, rather than CS8. We have revised the details in our records to reflect this more accurate staging in Table 1. We have included sections of this particular specimen for Figures for reviewer 1. Despite exhaustive sectioning until the sample was depleted, the heart structure was not located. The developmental stage of the specimen seems comparable to that of a mouse embryo at approximately embryonic day 7.5, evidenced by what appears to be a caudal neuropore. In addition, we observed surrounding blood vessels expressing PECAM, which contained nucleated red blood cells, but these did not exhibit Prox1 expression.

Figure for reviewer 1. Prox1 Expression Pattern in a CS9-10 Human Embryo.

Cross-section of a CS9-10 human embryo. Immunostaining for PECAM and Prox1.

The authors commented that CS11 lymphatic vessels were not identified in the vein. Was there any indication of LECs outside the vein? Could the authors include images of this stage?

Response:

The CS11 embryo is depicted in Supplemental Figure 2A-C'. In this section, identification of one side of

Revision Plan

the precardinal vein was possible. Furthermore, formation of the pharyngeal arch was observed. Prox1 expression was absent in the precardinal vein at this stage.

For embryos at CS12, it would be insightful to know the proportion of LECs versus VECs within the vein, the quantity of LECs outside the veins, and whether there was section-dependent variability in these observations.

Response:

For a single section, the numerical data for the right and left anterior cardinal veins were averaged. This process was repeated and the results were then averaged across two sections.

1. The proportion of LECs to VECs within the vein.

On average, there were 18.25 nuclei per cross-section of the CV; of these, 4.5 were Prox1⁻/PECAM⁺ blood endothelial cells (BECs), and 13.75 were Prox1⁺/PECAM⁺ LECs. Therefore, BECs constituted 24.7%, and LECs constituted 75.3%.

2. The number of LECs outside the veins.

There were an average of 9.75 Prox1⁺/PECAM⁺ cells located externally to the CV."

This point is described in Figure 1 legends as follows.

On average, there were 18.25 nuclei per cross-section of the CV; of these, 4.5 were Prox1⁻/PECAM⁺ blood endothelial cells (BECs), and 13.75 were Prox1⁺/PECAM⁺ LECs. Therefore, BECs constituted 24.7%, and LECs constituted 75.3%. There were an average of 9.75 Prox1⁺/PECAM⁺ cells located externally to the CV. (Page 11, lines 486-490)

It would be helpful if Table 1, "Information of human embryos and fetuses", could be complemented with a summary of the main findings at each stage, including which markers LECs expressed and their distribution.

To strengthen the assertion that this study provides unique insights compared to those of mice, a schematic summarizing the similarities and differences between mouse and human observations should be included.

Response:

We have enriched the information presented in Table 1 and introduced Figure 5 as a new comprehensive illustration. Figure 5 provides a comparative analysis of lymphatic vessel development between mice and humans, with a particular emphasis on the early stages of development, meticulously summarizing the alterations in lymphatic marker expression at specific stages.

The authors mentioned differences in lymphatic markers at various regions of the embryo and different developmental stages. It is essential to clarify whether all regions express the same markers at the latest developmental stage.

Response:

We have added immunostaining for Podoplanin and LYVE1 at GW9 as Supplemental Figure 4X-Y'''''. This demonstrates the expression of Podoplanin and LYVE1 in lymphatic vessels of the lung, heart, kidney, mesentery, intestinal wall, and lower jaw. This information regarding the expression of LYVE1

Revision Plan

and PDPN has also been incorporated into the main body of the text under the section of ‘The Development of Lymphatic Vessels Varies Among Organs’.

A discussion of the limitations of analysing embryos from abnormal pregnancies is necessary. In addition to the determined lack of chromosomal abnormalities, it is crucial to consider phenotypical and morphological integrity. The authors should address the possibility of developmental defects and mutations causing abnormalities in the lymphatic vessels.

Response:

In the "Tissue Collection and Ethical Considerations" section of the Materials and Methods, we have addressed the possibility that developmental defects and mutations may cause abnormalities in the lymphatic vessels.

This is depicted as follows:

Detailed information regarding each sample is presented in Table 1. The sex of each sample was not determined, with the exception of one case of miscarriage. In this particular case, chromosomal analysis verified the absence of any karyotypic abnormalities. There were no malformations observed in any of the embryos or fetuses. Nevertheless, for the remaining embryos, there is a possibility that developmental defects or mutations could lead to abnormalities in the lymphatic vessels. (Page 7, lines 340-346)

Minor comments

In the abstract, the authors refer to lymphatic malformations as a specific type of lymphatic disease. We recommend acknowledging the broader implications of this study beyond such specific cases.

Response:

We have modified the concluding paragraph of the Abstract to reflect a more expansive and encompassing narrative as follows.

Our research clarifies the early development of human lymphatic vessels, contributing to a better understanding of the evolution and phylogenetic relationships of lymphatic systems, and enriching our knowledge of the role of lymphatics in various human diseases. (Page 2, lines 58-60)

The term "lymph-related disease" should be clarified for better understanding.

Response:

To make it clearer, we have modified the last paragraph of the Introduction that includes 'lymph-related disease' as follows.

Our research offers essential insights into the evolution and phylogeny of lymphatic vessels, and may also illuminate the pathogenesis of lymphatic-related diseases, which include lymphedema, obesity, cardiovascular disorders, Crohn's disease, and congenital lymphatic disease, such as lymphatic malformation. (Page 3, lines 127-131)

Revision Plan

Figure 3S shows kidney samples, not the myocardium or endocardium, as indicated.

Response:

No, it is correct. Figure 3P-S represents the heart, which is surrounded by the lungs on both sides. Figure 3S depicts the endocardium, indicating that lymphatic vessels are not present within the endocardial layer.

Reviewer #1 (Significance (Required)):

This study largely reaffirms the existing knowledge from mouse models and previous human data. Given the absence of a cure for lymphatic diseases, gaining a deeper understanding of how lymphatic vessels develop in humans could serve as a crucial stepping stone in this field of research.

Reviewer #2 (Evidence, reproducibility and clarity (Required)):

This study by Yamaguchi et al., explores the progression of lymphatic vessel growth in different stages of human embryos. They also try to identify the origin of the lymphatic vessels in different organs. The study first shows that lymphatic endothelial cells (LECs) first show up in the anterior cardinal veins (ACVs) of CS12 in human embryos, which is similar to what is known to occur in mouse embryos. They also checked whether the PROX1+ LECs of the heart are derived from Flk1+/Isl+/PECAM- cells. However, Flk1+/Isl+/PECAM- cells do not co-express PROX1. These results suggest that in human embryos LECs originate from the ACVs. The authors then identify that lympho-venous valves formed between lymph sacs and the cardinal veins at around Carnegie Stage (CS)18. The valves have showed obvious bicuspid shape at Gestational week (GW)9. Finally, the authors demonstrate that the development of lymphatic vessels happens at different time points in various organs. At CS16, lymphatic vessels and LECs can be detected in the lower jaw, heart and the lungs; mesenteric and intestinal lymphatic vessels can be detected between CS17 and CS18; kidney lymphatic vessels can be found at CS23; At GW9, the lymphatic vessels are observed around the aorta, which may combine to form the future thoracic duct. Together, this informative study sheds light on the progression of lymphatic vasculatures during embryonic stage in humans.

This study has many strengths, in addition to some areas that if addressed, would further increase the impact of the findings. These include:

1. Since immunostaining is the major method that the authors have used for their work, they could use positive and negative controls (secondary antibody only or IgG control) for different antibodies. The authors can also show some Isl1 and Flk1 staining in GW9 fetus or adult tissue, like PROX1 or LYVE1 in Supplemental figure 1.

Response:

We have introduced new Supplemental Figures 1I-N. Included are negative controls for fluorescent staining with only the secondary antibody (Supplemental Figure 1I''''') and for DAB staining with only the secondary antibody (Supplemental Figure 1J-L). Furthermore, we have added images showing Flk1

Revision Plan

staining within lymph sacs (Supplemental Figure 1M) and Isl1 staining (Supplemental Figure 1N). Flk1 expression was confirmed in the lymph sacs; however, Isl1 expression was not observed.

The description regarding the negative controls is as follows.

Additionally, the specificity of the staining was confirmed with controls using only the secondary antibodies (Supplemental Figure 1I-L). (Page 3, lines 146-147)

The description regarding Flk1 and Isl1 in the lymph sac is as follows.

Additionally, at GW9, Flk1 expression was detected in the cervical lymph sac, but Isl1 expression was not (Supplemental Figure 1M and N). (Page 4, lines 186-187)

2. Figures 1 F-H, S' and S'', U', U'', and U''' are hard to appreciate. Can the authors offer higher quality images or show some confocal images?

Response:

In response to the reviewer's comments, we conducted several trials to improve image quality. However, due to fixation issues, we were unable to enhance the quality beyond the original for the CS12 specimen. Therefore, all images except those of VEGFR3 have been left unchanged. It is possible that the quality appeared reduced in the initial submission due to compression, making them difficult to view. We will resubmit without reducing the image quality as much as possible and ask for your understanding in this matter. Additionally, the CS12 specimen was very small, and there was a limited number of sections available, making further attempts challenging. This is also a limitation of research using human embryos. Regarding Figure 1R-U''', we have revised and replaced the images, although the quality has not significantly changed. We believe this may also be due to the compression of the image quality at the time of submission. There is no change in the conclusions drawn.

3. According to the author's previous publications (ref 17 and 30) and literature (ref 31), Flk1+/Isl1+/PECAM- cells differentiate into LECs. However, in this work they did not observe any PROX1+/Isl1+ cells at CS13 and CS14. I am curious to know if they found any PROX1+/Isl1+ cells at later time points such as GW9.

Response:

Isl1 is posited to be an early transcription factor that directs the differentiation of undifferentiated mesodermal cells towards a cardiac lineage. Our prior research utilizing tamoxifen-inducible mice indicated that a cohort of cells expressing Isl1 at a defined interval (E6.5 to E9.5 in mice) contributes to the formation of lymphatic structures in the head, neck, mediastinum, and heart before subsequently losing this expression (Maruyama et al., eLife, 2022). However, in human studies, it is not possible to trace the lineage and differentiation trajectories of *Isl1*⁺ cells. Consequently, we anticipated finding LECs that initially express Isl1 in the embryonic stage, with this expression diminishing as development ensues. Nevertheless, such cell groups were not observed in human embryos. In mice, our search for cells concurrently expressing Isl1, Prox1, Flk1, or PECAM from E9.0 to E11.5 (referenced in Maruyama et al., eLife, 2022, Supplemental Figure 3) also yielded no such populations. This evidence suggests that Isl1 protein expression in the cardiac pharyngeal mesoderm likely ceases during the differentiation into lymphatic endothelium. Given the hypothesis that Isl1+/Prox1+ LECs might exist at an earlier

developmental stage, we examined specimens from CS16, 17, and 18 for the presence of such LECs but to no avail. This investigation has been documented as Supplemental Figure 3Q-S for the CS16 sample. With the GW9 sample, due to its substantial size, we initially conducted a DAB staining search for lumen structures that might express Is11. However, no such structures were identified. Moreover, despite conducting triple immunostaining for PECAM, Is11, and Prox1, we were unable to locate any LECs or lymphatic vessels expressing Is11.

The description regarding Is11 and Prox1 expression for CS16 and GW9 is as follows:

At CS16, cells co-expressing Prox1 and Is11 were not observed in the lower jaw or the cardiac outflow tract regions (Supplemental Figure 3Q-S'''). Additionally, at GW9, Flk1 expression was detected in the cervical lymph sac, but Is11 expression was not (Supplemental Figure 1M and N). (Page 4, lines 184-187)

For the GW9 stage, we have provided images of lymphatic vessels in the lung and heart stained with PECAM, Is11, and Prox1 as a Figure for the reviewer's consideration.

Figure for reviewer 2. Is11 is not expressed in GW9 lymphatic vessels.

Fluorescent immunostaining of PECAM, Prox1, and VEGFR3 was conducted at GW 9 fetuses. Scale bars 100 μ m.

4. Figure 3 N and O show comparable VEGFR3+PROX1+ cell numbers in different time points, however it shows increased VEGFR3+PROX1+ vessel numbers. If so, do LECs become more elongated and form the vessel-like structures?

Response:

In our previous findings (Maruyama et al., Dev bio, 2019, Maruyama et al., iScience, 2021), we documented that surrounding the heart, LECs progressively interconnect to form a reticular network, which is subsequently remodeled into more substantial lumen-bearing vessels. This sequence appears to be conserved in humans, with LECs initially presenting as solitary entities that gradually interlace into a network. Presumably, a portion of this network is then streamlined, giving rise to increasingly luminal structures. Therefore, while the count of LECs remains constant, there is an augmentation in the number of defined luminal vessels. This observation has been depicted as follows.

Revision Plan

Throughout this process, the initially mesh-like capillary lymphatics undergo progressive remodeling to establish lumen-bearing vessels. Consequently, while the density of LECs per unit area remains relatively stable, there is an increase in the number of lymphatic vessels possessing distinct luminal structures (Figure 3N and O). (Page 5, lines 220-223)

5. The authors have mentioned that the staging of the embryos and fetuses was done by Carnegie stage and clinical information. The authors should offer more detailed information about those embryos and fetuses. For example, crown-rump length, menstrual weeks, craniofacial features etc. This information will be useful for other researchers in this field.

Reply:

We have substantially expanded the data presented in Table 1 regarding embryos and fetuses. For specimens dating back over 15 years, some lacked echo graphic details. In those instances, we estimated the developmental stage by integrating available data, such as the date of the last menstrual period or morphological features of the fetus. For a case initially assessed as CS 8, which had no recorded cardiac activity in the preceding week, a subsequent ultrasound noted a heartbeat. Considering this alongside the specimen's size, we revised the estimated stage to CS 9-10, correlating with the onset of heart formation. Despite exhaustive sectioning of this particular embryo until the samples were depleted, the heart structure remained undetected. Nevertheless, taking into account morphological observations, such as an open neural tube, the stage was adjudged to be CS9-10. Furthermore, for ectopic pregnancies, which frequently necessitated emergency surgeries due to symptoms like abdominal pain or bleeding, preoperative embryonic data was often unavailable.

Reviewer #2 (Significance (Required)):

Strengths: Very informative results for human embryonic lymphatic development. They have performed the experiments at various developmental stages.

Limitations: Image quality need to be improved. Many high magnification images are not clear. Human samples come from certain diseases, which might have affected the embryo's development.

Advance: this study clarified the process of early lymphatic vessel formation in human embryos.

Audience: clinical and basic science in developmental biology and lymphatic biology.

Reviewer expertise: lymphatic development, lymphatic biology, vascular biology.

Revision Plan

4. Description of analyses that authors prefer not to carry out

Since I have addressed most of the comments, they are documented in Section 3.

Dear Kazuaki,

Congratulations on a great revision! Overall, the referees have been positive. One referee has expressed one additional concern that we ask you to address either experimentally (additional staining) or by adding to the discussion.

When you submit your revised version, please also take care of the following editorial items and add this also to your point-by-point response:

1. Please provide an author checklist found online.
2. Please remove the main figures from the manuscript text and upload them as high resolution figure files; legends should stay in the manuscript. Supplemental figures should be renamed "Figure EV1" etc. and also uploaded as individual high resolution figure files. Legends should be added to the manuscript, under the heading "Expanded View Figures".
3. Please provide the institutional emails for corresponding authors and please switch the order in your account in eJP to institutional email and primary email address.
4. Please update the data availability section title to: "Data Availability" and the text to "This study includes no data deposited in external repositories."
5. Please include all relevant funders to this study in our eJP system.
6. Please remove the author contribution section from the main manuscript text.
7. Please review our new policy on conflict of interests on the EMBO author guide website and update the title of this section to: "Disclosure and competing interests statement".
8. Please correct the reference format to alphabetical order and 10 author names listed before et al.
9. We do not allow the phrase "data not shown" in our publications. Please remove these from p7
10. We include a synopsis of the paper (see <http://emboj.embopress.org/>). Please provide me with a general summary statement and 3-5 bullet points that capture the key findings of the paper.
11. We also need a summary figure for the synopsis. The size should be 550 wide by 200-440 high (pixels). You can also use something from the figures if that is easier.
12. We require that all figures be referred to in the main manuscript. Please include a call out to Figure 5 in the manuscript.
13. Please remove "classification" information from the main manuscript.
14. Please correct the order of manuscript sections to: Abstract, Introduction, Results, Discussion, Materials and Methods, Acknowledgements, Disclosure and competing interests statement, References, Figure legends, Tables and their legends, Expanded View Figure legends
15. Please define the error bars in the legends of figures 3n-o, 4i-j, w-x.
16. Please add a scale bar for figure 4e-h.
17. Please add a scale bar and definition for figure 1c-e, 1g-i, 2c-d, 2k-l, 3c-e, 3l-m.
18. Please define the red arrows in the legend of figure 1k.
19. Please also update the ethical section as we discussed, answering the questions from our expert in this area.

Thank you for the opportunity to consider your work for publication. I look forward to your revision.

Kind regards,
Kelly

Kelly M Anderson, PhD
Editor, The EMBO Journal

k.anderson@embojournal.org

Please submit your revised manuscript using the following link:

Link Not Available

Referee #1:

The authors have addressed most of the previous comments.

I only have one more major comment:

The authors have described in both the abstract and summary sections that each organ could have different lymphatic origins. I suggest the authors perform more staining to support this conclusion. For example, whether those small LEC populations are more proliferative, express progenitor cell markers, or connect to the veins.

Additional suggestion:

Due to all the data being descriptive results, I suggest the authors submit this paper to the EMBO report or Life Science Alliance.

Referee #2:

The authors have addressed all the revision points, I have no further comments.

Rev_Com_number: RC-2023-02149

New_manu_number: EMBOJ-2023-116122

Corr_author: Maruyama

Title: Lymphatic vessel development in human embryos

EMBOJ-2023-116122

First of all, we extend our sincerest thanks for accepting to review our work amidst your busy schedule.

Referee #1:

The authors have addressed most of the previous comments.

I only have one more major comment:

The authors have described in both the abstract and summary sections that each organ could have different lymphatic origins. I suggest the authors perform more staining to support this conclusion. For example, whether those small LEC populations are more proliferative, express progenitor cell markers, or connect to the veins.

Response:

The areas highlighted in your comments are very important and have also been a focus of our attention. With respect to the trunk region, we have included Figure 1 based on strong evidence from murine studies that, alongside Prox1, COUP-TF2 is essential for the emergence of LECs from the cardinal vein. In contrast, the LECs in the head, neck, and facial areas (originate from cardiopharyngeal mesoderm) appear to originate not from the sprouting of existing veins but through a process of lymphvasculogenesis. While we have explored the potential of capillary budding as a source and conducted analyses, our findings did not support this (Maruyama et al., 2019, 2021, 2022).

To elucidate the molecular mechanisms by which LECs from the cardiopharyngeal mesoderm differentiate in the head and neck regions, we reanalyzed available single-cell data and gathered new data for analysis. This has progressively shed light on the differentiation of head and neck region LECs in mice. We then conducted immunostaining for transcription factors anticipated to be pivotal in early human differentiation, but, regrettably, the antibodies did not yield results, leaving this theory unverified. This area warrants further study.

As for proliferative activity, Ki67 immunostaining was conducted, and the results are presented in the newly added Figure EV5. We analyzed samples from CS13 and 16. At CS13, a higher proportion of Ki67-positive LECs were observed in extra-CV regions compared to CV. At CS16, despite a reduction in overall embryonic Ki67 activity, 31% of LECs in the lower jaw remained Ki67 positive, and about 22.7% in the lymph sacs were Ki67 positive.

The differential rates of lymphatic vessel development across organs could be attributable

to distinct developmental origins or may suggest an interaction between organogenesis and lymphangiogenesis (Liu et al., Nature, 2020). Notably, our research provides the initial evidence of significant temporal disparities in lymphatic vessel development between humans and mice. We conclude that ongoing analysis into the signaling pathways and origins is imperative in future.

These findings are depicted in the main manuscript as follows:

Proliferative activity of LECs during the organogenesis period

To examine LEC proliferation activity, we conducted Ki67, Prox1, and PECAM staining in specimens from CS 13 and 16. At CS13, CV showed a Ki67 positivity in 19.6% of LECs, while extra-CV LECs exhibited a 43.6%. At CS16, lymph sacs displayed a Ki67 positivity rate of 22.7%, and the LECs in the lower jaw were positive at a rate of 31% (Figure EV5A-F”). (p6, lines 239-243)

Additionally, there is a word limit for the Abstract, so it has been significantly shortened from original version.

Additional suggestion:

Due to all the data being descriptive results, I suggest the authors submit this paper to the EMBO report or Life Science Alliance.

Response:

Thank you for your suggestion.

As previously stated, we recognize the significance of molecular-level analysis and are diligently pursuing it. However, it appears that additional time is required to integrate these insights into phenomena at the organismal level. However, this work addresses a broad concept relevant to all of us as humans, and we anticipate it will appeal to a broad readership. With this in mind, we wish to submit our work for publication to “The EMBO journal”. We sincerely hope for your understanding and extend our heartfelt thanks in advance.

Dear Kazuaki,

Congratulations on an excellent manuscript, I am pleased to inform you that your manuscript has been accepted for publication in The EMBO Journal. Thank you for your comprehensive response to the referee concerns and for providing detailed source data. It has been a pleasure to work with you to get this to the acceptance stage.

I will begin the final checks on your manuscript before submitting to the publisher next week. Once at the publisher, it will take about 3 weeks for your manuscript to be published online. As a reminder, the entire review process, including referee concerns and your point-by-point response, will be available to readers.

I will be in touch throughout the final editorial process until publication. In the meantime, I hope you find time to celebrate!

Kind regards,
Kelly

Kelly M Anderson, PhD
Editor, The EMBO Journal
k.anderson@embojournal.org

Rev_Com_number: RC-2023-02149
New_manu_number: EMBOJ-2023-116122R
Corr_author: Maruyama
Title: Lymphatic vessel development in human embryos